# Reversion analysis reveals the in vivo immunogenicity of a poorly MHC I-binding cancer neoepitope

Hakimeh Ebrahimi-Nik [1,8], Marmar Moussa[1,10], Ryan P. Englander [1,10], Summit Singhaviranon[1], Justine Michaux[2,3], HuiSong Pak[2,3], Hiroko Miyadera [4,5], William L. Corwin[1,9], Grant L. J. Keller [6], Adam T. Hagymasi[1], Tatiana V. Shcheglova[1], George Coukos [2,3], Brian M. Baker [6], Ion I. Mandoiu [7], Michal Bassani-Sternberg [2,3] & Pramod K. Srivastava [1✉]

High-affinity MHC I-peptide interactions are considered essential for immunogenicity. However, some neo-epitopes with low affinity for MHC I have been reported to elicit CD8 T cell dependent tumor rejection in immunization-challenge studies. Here we show in a mouse model that a neo-epitope that poorly binds to MHC I is able to enhance the immunogenicity of a tumor in the absence of immunization. Fibrosarcoma cells with a naturally occurring mutation are edited to their wild type counterpart; the mutation is then re-introduced in order to obtain a cell line that is genetically identical to the wild type except for the neo-epitope-encoding mutation. Upon transplantation into syngeneic mice, all three cell lines form tumors that are infiltrated with activated T cells. However, lymphocytes from the two tumors that harbor the mutation show significantly stronger transcriptional signatures of cytotoxicity and TCR engagement, and induce greater breadth of TCR reactivity than those of the wild type tumors. Structural modeling of the neo-epitope peptide/MHC I pairs suggests increased hydrophobicity of the neo-epitope surface, consistent with higher TCR reactivity. These results confirm the in vivo immunogenicity of low affinity or 'non-binding' epitopes that do not follow the canonical concept of MHC I-peptide recognition.

[1] Department of Immunology and Carole and Ray Neag Comprehensive Cancer Center, University of Connecticut School of Medicine, Farmington, CT, USA. [2] Ludwig Institute for Cancer Research, University of Lausanne, Lausanne, Switzerland. [3] Department of Oncology, Centre hospitalier universitaire vaudois (CHUV), Lausanne, Switzerland. [4] Department of Medical Genetics, Faculty of Medicine, University of Tsukuba, Ibaraki, Japan. [5] Genome Medical Science Project, National Center for Global Health and Medicine, Chiba, Japan. [6] Department of Chemistry and Biochemistry and Harper Cancer Research Institute, University of Notre Dame, Notre Dame, IN, USA. [7] Department of Computer Sciences, University of Connecticut School of Engineering, Storrs, CT, USA. [8] Present address: Broad Institute of MIT and Harvard, 105 Broadway, Cambridge, MA, USA. [9] Present address: Arvinas, 5 science park, 395 Winchester Ave, New Haven, CT, USA. [10] These authors contributed equally: Marmar Moussa, Ryan P. Englander. ✉email: Srivastava@uchc.edu

Antigen presentation by MHC molecules is fundamental to adaptive immunity. In the case of MHC I molecules, such presentation involves a complex series of steps that result in the proteolytic processing of whole or partially synthesized proteins, chaperoning of the peptides through the cytosol and the endoplasmic reticulum, and their rendezvous with MHC I molecules into a tri-molecular MHC I-β2 microglobulin-peptide (pMHC) complex[1]. Based on extensive analyses of peptides recognized by mouse and human T cells against viral antigens, it has been clear that a high affinity (IC50 values <500 nM, but preferably <50 nM) of peptides for MHC I is essential for antigen presentation[2]. This premise has been abundantly validated in its ability to predict the epitopes that can elicit a CD8+ T cell response measurable in vitro[3].

Subsequent to the advances in our ability to identify somatic mutations in cancers, identification of epitopes that can act as cancer vaccines has become a large area of inquiry. Since affinity of peptides to MHC I has withstood the test of time as a key criterion for predicting immunogenicity, this has been applied to the discovery of cancer neoepitopes as well, and a number of high affinity neoepitopes that elicit tumor rejection as well as CD8 T cell responses measurable in vitro, have been identified[4–6]. A measurable CD8 response is often considered a valid surrogate for tumor rejection, and several neoepitopes, which have a high affinity for MHC I, and elicit CD8 T cell response have been identified[7–9]. Indeed, high affinity of a peptide for MHC I has become so entrenched in immunological thought that peptides with a low affinity (IC50 of >500 nM) are routinely excluded from consideration as candidates for vaccines, and are even often referred to as "non binders" to reinforce their irrelevance.

A small number of recent reports have examined the question of immunogenicity of mouse cancer neoepitopes from a vantage point agnostic to peptide-MHC I affinity. Such studies have reported a number of neoepitopes which bind MHC I with low affinity, and mediate CD8-dependent tumor rejection[10,11]. At the same time, two retrospective human studies analyzing the genomic and clinical outcome data from nearly 7,000 patients with 27 cancer types, have shown that better clinical outcomes and T cell infiltration of tumors are associated with the presence of cancer neoepitopes with low affinities for HLA I molecules, and not with the presence of high affinity HLA I-binding neoepitopes[12,13].

Consistent with the lack of association between high affinity of neoepitope to MHC I and anti-tumor activity, all high affinity binding neoepitopes failed to elicit tumor rejection in a mouse model of ovarian cancer[14]. Human clinical trials with high affinity neoepitopes have also failed to elicit significant CD8 T cell responses even when high affinity MHC I binding algorithms were used to predict the immunizing neoepitopes[15–17]. Such clinical trials have also not shown convincing evidence of anti-tumor activity of the immunizing neoepitopes.

Since the ability of a neoepitope with poor affinity for MHC I to mediate CD8-dependent tumor rejection runs contrary to our dominant conception of MHC I-peptide interaction, it deserves critical scrutiny.

Here we show that the presence or absence of a low affinity MHC I-binding neoepitope in the tumor influences the spontaneous immunogenicity of a tumor in vivo. Upon transplantation, a mouse fibrosarcoma cell line, bearing a mutation known for encoding an MHC I 'non-binder' neoepitope, becomes less immunogenic when the sequence is reverted to the wild type allele and regains the original T cell activating capacity when the mutation is re-introduced. These experiments clearly demonstrate that a single MHC I 'non-binder' neo-epitope drives spontaneous immunogenicity of the fibrosarcoma and thereby challenge the current view of MHC I affinity determining the tumor immune response.

## Results

**Definition of the neoepitope Ccdc85c^MUT.** The *Ccdc85c* gene encodes a gap junction protein expressed mostly in the brain, colon, lung, kidney and testes in adult mice. The protein has no known oncogenic (driver) function. A non-synonymous (leucine to phenylalanine, Chromosome 12-108221754) somatic SNV in *Ccdc85c* was detected in the BALB/cJ Meth A fibrosarcoma (Fig. 1a). (See Duan et al. Supplementary Table S1 for a list of all mutations and predicted neoepitopes of the Meth A sarcoma).

The mutation is heterozygous and the un-mutated as well as the mutated reads are detected in the transcripts. BALB/cJ bone marrow derived dendritic cells (BMDCs) pulsed with an 18-mer peptide with the mutant amino acid near the center (DPSSTYIRPFETKVKLLD) or un-mutated peptide (DPSSTYIR-PLETKVKLLD), were used to immunize BALB/cJ mice. All mice were challenged with the Meth A cells, and tumor rejection was monitored (Fig. 1b upper panel). Immunization of BALB/cJ mice with the mutated Ccdc85c 18-mer elicited potent tumor rejection or control in all mice, while the un-mutated peptide failed to elicit protection (Fig. 1b lower panel). The anti-tumor activity of Ccdc85c^MUT was abrogated by depleting the mice of CD8 cells by treating the mice with the anti-CD8 antibody but not by a control antibody during the priming phase as previously described[11].

Various truncated versions of the 18-mer peptide as indicated in Fig. 1c (and Supplementary Fig. 1) were similarly tested for tumor rejection. A tumor rejection score (TRS) (with a maximum score of 5 indicating near 100% tumor rejection) was used to quantitate the extent of tumor rejection as described in Methods. The 18-mer peptide elicited a perfect 5.0 TRS score (Fig. 1c). The most and the least effective peptides along with their TRS scores are shown in Fig. 1d. Since the 10 amino acid peptide YIRPFETKVK was the shortest peptide active in tumor rejection, we consider this the precise epitope.

Evidence of presentation of YIRPFETKVK was sought by analyzing the peptides eluted from MHC I molecules purified from the Meth A cells by mass spectrometry (MS), as described in Methods. No Ccdc85c-derived peptides were detected, as expected from the low abundance of expression of this protein. In order to identify the precise peptide derived from the mutant Ccdc85c that could be cross-presented by the DCs, BMDCs were pulsed in vitro with the 18-mer peptide as previously described[11]. The BMDCs were extensively washed and MHC I molecules eluted. Targeted-MS analysis of the eluted peptides in the presence of spiked-in heavy labeled synthetic peptides showed the presence of two Ccdc85c-derived peptides TYIRPFETKVK and YIRPFETKVK (Fig. 1e). These two peptides detected by cross-presentation of the 18-mer peptide were identical to the two truncated versions of the 18-mer peptide that were observed to be the most effective in tumor rejection (Fig. 1d). These peptides had very low or undetectable predicted as well as measured affinities for $K^d$, $D^d$ and $L^d$ as shown for $K^d$ in Fig. 1f. With such low affinities, these neoepitopes would normally be considered non-binders.

The 18-mer sequence was queried for the presence of predicted $K^d$, $D^d$ or $L^d$-binding peptides. No $D^d$ or $L^d$-binding peptides were predicted; three peptides were predicted to bind $K^d$ albeit with poor affinity (IC$_{50}$ values between 692 and 864 nM) (Fig. 1f). Ironically, none of these three peptides were detected by MS among peptides eluted from MHC I of BMDCs pulsed with the 18-mer long peptide.

In order to determine if any peptides within Ccdc85c^MUT could be presented by MHC II molecules, we analyzed the interaction of *H2-A^d* and *H2-E^d* with TYIRPFETKVK, YIRP-FETKVK and IRPFETKVK as well as their wild type counterparts, using a cell-surface density assay. In this assay, β-chains of H2-Ab1^d or H2-Eb1^d are expressed in fusion with the peptide of

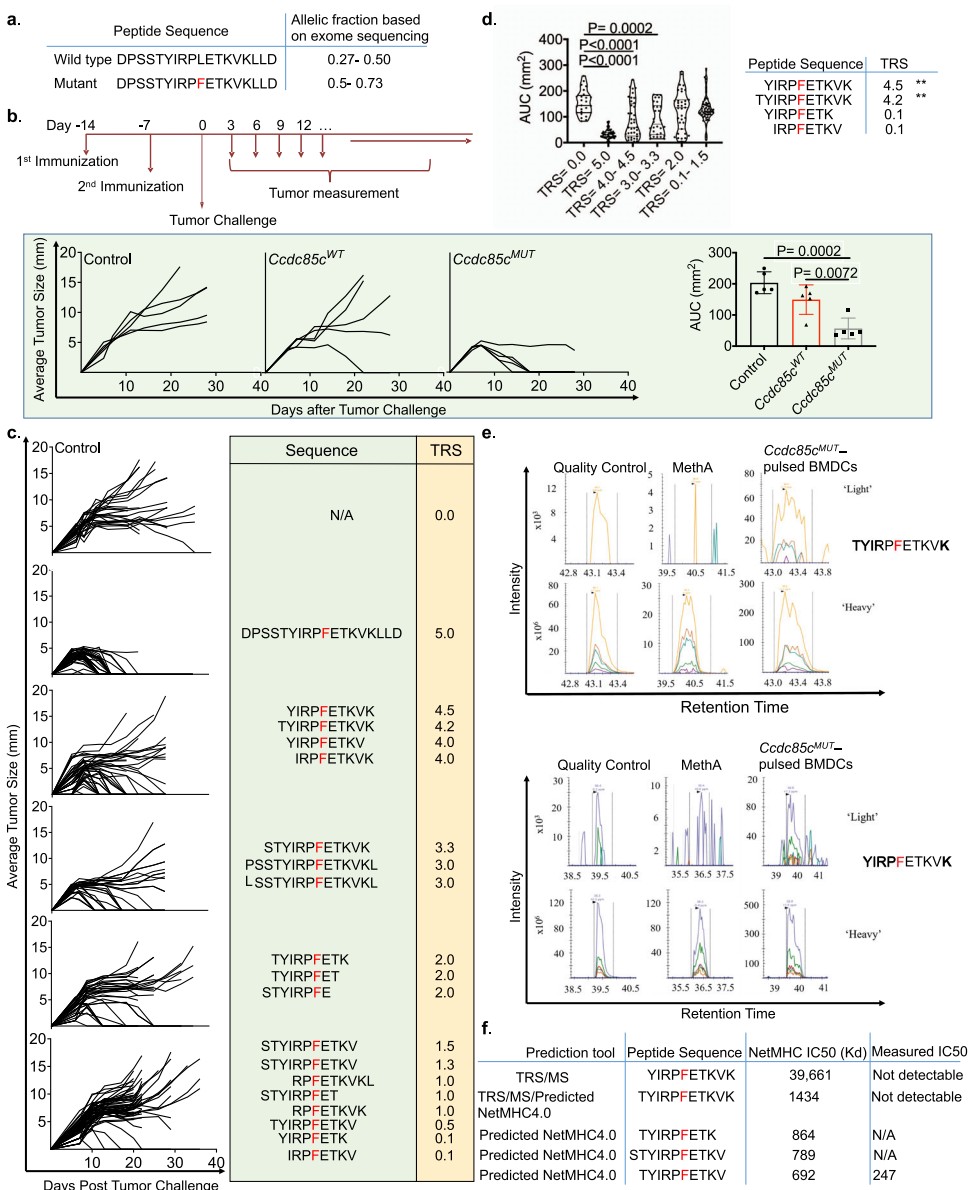

**Fig. 1 Definition of the precise neoepitope of Ccdc85c^MUT that mediates tumor rejection. a** The sequences of the 18-mer wild type and mutant peptides derived from *Ccdc85c* gene as well as their corresponding allelic fractions (the number of mutant/normal reads divided by the total number of reads (coverage) at a specific genomic position) are shown. **b** The top panel shows a schematic diagram of immunization and tumor challenge in BALB/cJ mice. The bottom panel (left) shows tumor growth in BALB/cJ mice immunized with Ccdc85c^MUT or Ccdc85c^WT and challenged with Meth A as described in Methods. Each line represents tumor growth in a single mouse ($n = 5$ mice per group). AUC for each group is plotted in the panel on the right. Data are presented as mean ± SD. *P* values were calculated using 1-way ANOVA test adjusted for multiple comparisons (Tukey's multiple comparison test). **c** Several truncated versions of the 18-mer *Ccdc85c^MUT* peptide were tested in tumor rejection assay. BALB/cJ mice were immunized and tumor challenged. Each line represents tumor growth in a single mouse. Although mice were immunized with individual peptides, the data for multiple peptides are grouped into one with the composition of the peptides shown on the right. The tumor rejection data for individual peptides are shown in Supplementary Fig. 1. Tumor rejection score (TRS) for each neoepitope is shown in the yellow box, where five represents a complete tumor protection and zero means no tumor rejection. **d** On the left panel, total Area Under the Curve (AUC) scores for each group in B are plotted. Each bar shows the average total AUC score for the indicated group (TRS = 5; $n = 35$, TRS = 4–4.5; $n = 40$, TRS = 3–3.3; $n = 25$, TRS = 2; $n = 30$, TRS = 0.1–1.5; $n = 60$). Error bars represent standard deviation (SD). The P values corresponding to the comparison of TRS = 0 with TRSs 5.0, 4.0–4.5 and 3.0–3.3 were respectively <0.0001, <0.0001 and 0.0002. *P* values were calculated using 1-way ANOVA test adjusted for multiple comparisons. On the right, peptides with the highest and the lowest TRS are shown. **e** Targeted MS-based detection of TYIRPFETKVK and YIRPFETKVK among MHC I peptides eluted from BMDCs pulsed with the 18-mer Ccdc85c^MUT. Heavy labeled synthetic peptides were spiked into the peptide samples; the labeled amino acid is marked with a bold character and the mutation is in red. Matched peak lists for the "heavy" and "light" ions were extracted and monitored, while only single charge y ions were plotted. See Methods for details. **f** Predicted (by NetMHC4.0) and measured IC50 values of the binding of candidate precise neoepitopes of Ccdc85c^MUT to K^d are shown. The candidate neoepitopes include those defined by tumor rejection and MS as in panels **c** and **e**. The other three candidate neoepitopes were predicted by NetMHC4.0 alone and were not active in tumor rejection. The affinities of the MS/TRS predicted neoepitopes were also measured for D^d and L^d; measured affinities were below the level of detection.

interest and the amount of cell-surface MHC II, as a measure of intrinsic stability of MHC II, is quantified in engineered conditions[18]. The measured cell-surface MHC density correlates well with the actual affinity of the peptide to MHC II. No significant difference was observed in binding of Ccdc85c$^{MUT}$ and Ccdc85c$^{WT}$ to H2-IA or H2-IE (Supplementary Table 1 and Supplementary Fig. 2a). Consistent with these findings, depletion of CD4 T cells in mice immunized with Ccdc85c$^{MUT}$ did not lead to reproducible and statistically significant abrogation of tumor rejection (Supplementary Fig. 2b). This was observed when CD4 depletion was carried out during the priming phase alone (as in Supplementary Fig. 2b), or throughout the entire experiment including the effector phase (i.e. post tumor challenge).

**Analysis of immune response against Ccdc85c$^{MUT}$.** BALB/c mice were immunized with 18-mer peptides containing Ccdc85c$^{MUT}$ as well as Ccdc85c$^{WT}$ as described in Methods. Spleens were harvested seven days after the second immunization and splenic CD8 T cells tested by ELIspot for interferon γ production in response to stimulation in vitro with the peptides used for immunization. Mice immunized with the wild type peptide showed no CD8 response while the mice immunized with the mutant peptide showed a very modest but statistically significant CD8 response (Supplementary Fig. 3). Since the CD8 response detected in vitro was so modest, CD8 response as detected in vivo in the tumor microenvironment became our focus.

Tumor microenvironment (TME) of Ccdc85c$^{MUT}$-immunized mice was examined using single cell RNA sequencing (scRNA seq). BMDCs- immunized mice were used as controls. As a peptide control, mice immunized with a neoepitope Alms1$^{MUT}$ were used. During comparison of the sequences of Meth A exomes with the normal BALB/cJ exomes, we identified Alms1$^{MUT}$ (LYLDSKSDTTV) which was also identified among the peptides eluted from MHC I molecules from BMDCs pulsed with the 18-mer Alms1$^{MUT}$ peptide. Although this neoepitope has a high affinity for a mouse MHC I K$^d$ (IC$_{50}$ 62.25 nM), and it can be processed and presented, immunization of mice with the 18-mer Alms1$^{MUT}$ failed to elicit tumor rejection (Supplementary Fig. 4). Hence, this peptide was chosen as a control peptide. Mice were immunized with Ccdc85c$^{MUT}$, Alms1$^{MUT}$ (peptide control) or BMDCs (control) as described in Methods, and were challenged with Meth A. RNA from CD45 + cells (estimated 15,925 cells for the four libraries after QC, with an average coverage of 32,663 reads per cell and median 1,339 genes per cell) was sequenced. Data from the four libraries were pooled and clustered based on the gene expression pattern of each as described in Methods. Annotation of the clusters was informed by both differentially expressed genes (DE Genes) and per cluster highly expressed genes identified by the TF-IDF analysis as described in Methods. Two major and distinct clusters were identified namely, myeloid and lymphoid. Upregulated genes used to identify the myeloid cluster included, but were not limited to: *Itgam* (CD11b), *Adgre1* (F4/80), *Arg1* (Arginase 1), *Nos2* (Nitric oxide synthase 2), *Ms4a4c* (Membrane-spanning 4-domains, subfamily A, member 4 C), *C1qa* (Complement component 1) were highly expressed in the myeloid cluster. Upregulated genes used to identify the lymphoid cluster included, but were not limited to: (*Cd3* (CD3), *Ptprcap* (CD45-AP), *Nkg7* (Protein NKG7), *Cd28* (CD28), *Gzma* (Granzyme a), *Prf1* (Perforin1) etc.) (Fig. 2a, left panel). The proportion of lymphoid versus myeloid compartments in the Ccdc85c$^{MUT}$ library was very different compared to the Alms1$^{MUT}$ or control groups. The Ccdc85c$^{MUT}$ library was mostly composed of lymphoid cells (62.68% lymphoid and 37.32% myeloid), while the control groups and mice immunized with Alms1$^{MUT}$ were mostly composed of

the myeloid compartment (~35% lymphoid and ~64% myeloid) (Fig. 2a, right panel). In order to study different cell types, lymphoid and myeloid clusters were re-clustered into 6 and 9 sub-clusters, respectively (Fig. 2a, bottom panel). The six identified lymphoid clusters were: CD4 T cells (CD4(1)), NKC(1), naive/early activated CD4 T cells (CD4(2), defined by a high expression of *Sell*, *Il7r*, *Tcf7* and *Ccr7* genes and low expression of *Il2ra* and lack of expression of effector and cytotoxicity genes), NKC(2) (less cytotoxic and active than NKC(1)), CD8 T cells (CD8) and proliferating CD4/CD8 T cells (Pr. CD4/CD8, defined by higher expression of *Stmn1* and *Mki67* genes and cell cycle gene expression analysis, described in Methods). The selected genes used as markers to annotate each lymphocyte cluster are listed in the summary heat map (Fig. 2b, right panel).

The majority of NK Cells (~80%) in the Ccdc85c$^{MUT}$ library were from the NKC(1) cluster which was the more cytotoxic and active cluster (defined by higher expression of *Cd44*, *Tnfa*, and *Il7r*), while, the fraction of active NK cells in other libraries was about 55%.

To pinpoint differences in T cells of the four libraries, clusters 1 and 5 (activated CD4 and CD8 T cells) were computationally pooled and the expression of cytotoxicity and other effector function genes were compared between libraries. Proliferating CD4/CD8 T were excluded from further analysis because the gene expression levels in these cells could be influenced by the cell cycle effect prominent in this cluster. Interestingly, Ccdc85c$^{MUT}$ library had the most contribution to the aforementioned pooled cells (31% Ccdc85c$^{MUT}$, 25% Alms1$^{MUT}$, 21% Ctlr1, 20% Ctrl2). Also, the normalized average gene expression (described in Methods) of cytotoxicity (*Gzmb*, *Prf*, and *Nkg7*) and other effector function (*Ifng*) genes were significantly higher in T cells derived from the Ccdc85c$^{MUT}$ library compared to the control or Alms1$^{MUT}$ libraries. Similarly, T cells of Ccdc85c$^{MUT}$ library had a significantly higher expression of genes involved in TCR engagement (*Nr4a1* and *Irf4*). A transcription factor involved in transcription of cytotoxicity genes, *Eomes*, had a significantly higher expression in T cells of Ccdc85c$^{MUT}$ library (Fig. 2c).

In the myeloid compartment, nine distinct clusters were identified. These are: macrophage1 (Mφ1), Mφ2 (defined by a moderate expression of *Arg1* and lower expression of *Cd302*, *Ccl5*, *Ccl8* and *C1qa*), monocyte1 (Mo1), Mφ3 (defined by a lower expression of *Ccl8* and a higher expression of *Ly6c*, *Cxcl9*, *Il1b*, *H2-Ab1*, *H2-DMb2*, *Mmp14*, and *Cd38*), Mφ4 (defined by a higher expression of *Nos2*, *Mrc1*, *Itgam*, *Pf4*, *C1qa*, *C1qb*, and *C1qc*), DC1, Mo2 (defined by a higher expression of *Itgax*, *Tlr7*, *Ace*, and *Adgre4*), neutrophil (Ne) and DC2 (defined by a higher expression of *Ccr7*, *Ccl5*, *Samsn1*, *Pcgf5*, *Gyg*, *Net1* and *Rabgap1l*) (Fig. 2b, left panel). The contribution of library Ccdc85c$^{MUT}$ to the most of myeloid clusters was minimal (ranging from 1.6% in Mφ3 to 13.4% in Mφ2). The only exception is for Mφ1 which Ccdc85c$^{MUT}$ library forms ~20% of this cluster (Supplementary Fig. 5).

**Mutation-reversion analysis of CD8 T cell immunogenicity of Ccdc85c$^{MUT}$.** The studies described above examined the immunogenicity of Ccdc85c$^{MUT}$ when administered as a vaccine. We aimed now to analyze the role of Ccdc85c$^{MUT}$ in the immunogenicity of the Meth A tumor itself, and in vivo. The broader objective was to test if a poorly MHC I-binding neoepitope residing within a tumor influences the CD8 immunogenicity of the tumor. CRISPR-guided gene editing was used to generate two variants of the Meth A. For purpose of this experiment, we refer to the original Meth A tumor with the endogenous Ccdc85c$^{MUT}$ as MUT1. Using CRISPR, the point mutation in Ccdc85c$^{MUT}$ was reversed back to its WT counterpart as described in Methods and

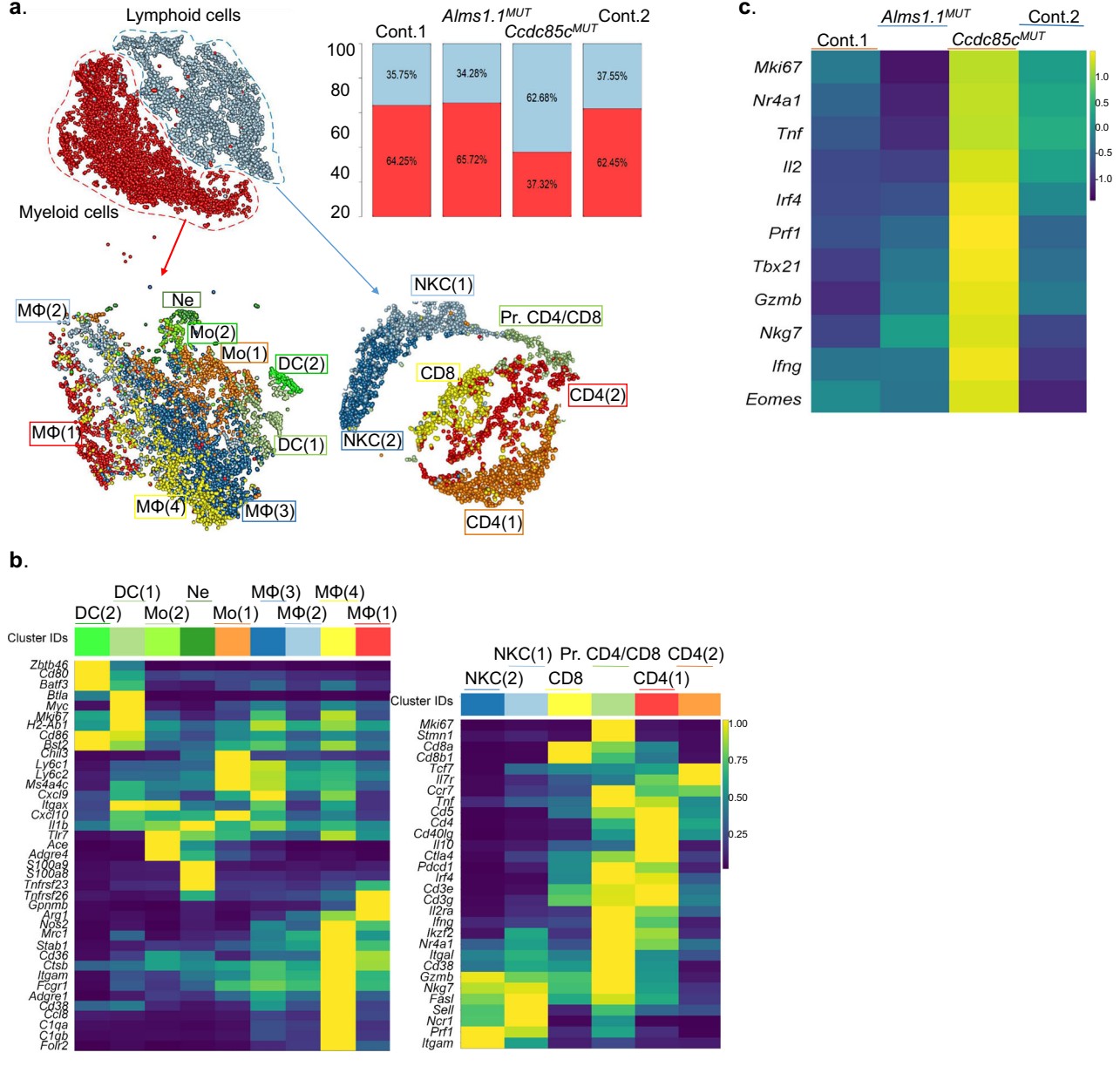

**Fig. 2 Immune response in mice immunized with Ccdc85c^MUT and control neoepitope Alms1.1^MUT. a** Tumors from mice immunized with Ccdc85c^MUT or Alms1.1^MUT as well as two groups of control mice immunized with BMDCs were harvested 10 days after tumor challenge. Tumor infiltrating CD45 + cells were sorted and analyzed by scRNA seq as described in Methods and Supplementary Fig 9b. Combined scRNA seq data from four libraries were analyzed. The t-SNE[34] plots of all intra-tumoral immune cells as well as myeloid and lymphoid compartments are shown in the top left panel. Myeloid and lymphoid composition for each library is shown on the right panel. The myeloid and lymphoid sub-clusters are shown in the bottom panels. See text for definition of each sub-cluster. **b** Heat maps of the indicated genes in the myeloid and lymphoid compartments are shown. The heat map genes were selected from the list of differentially expressed genes, genes with a high average TF-IDF score, and typical cell type markers. The average gene expression for each cluster was normalized by dividing it from the maximum value of each row (gene). **c** Summary heat map of selected genes associated with cytotoxicity for pooled activated T cells is shown. The heat map illustrates scaled (Z-scored) average gene expression by library.

Supplementary Fig. 6; this line with the WT sequence of *Ccdc85c* is referred to as a Revertant (REV). As shown in Supplementary Fig. 6b, the REV tumor grows significantly faster than the original MUT1 line. The point mutation in Ccdc85c^MUT was then re-introduced into the Revertant to generate the line MUT2 which recapitulates the original mutation as in Ccdc85c^MUT. Thus, MUT1 and MUT2 are identical tumors (except that MUT1 is heterozygous for the mutation while MUT2 is homozygous for it) and are different from REV only with respect to the mutation in Ccdc85c^MUT. Groups of mice were challenged with the three tumors individually (MUT1, REV or MUT2) and their TILs

analyzed by scRNA seq. RNA from CD45 + cells (estimated 11,961 cells for the three libraries before QC and 10,265 after QC, and with an average coverage of 61,371 reads per cell and median of 2,137 genes per cell) was sequenced. Data from three libraries were pooled and expression of the top average TF-IDF scoring genes was compared between the three libraries (Fig. 3a). CD8 T cells of all three libraries showed CD8 activation markers including but not limited to: *Cd69* and *Cd44* as activation markers, *Lamp1* (CD107a) as a measure of degranulation and *Tbx21* (Tbet), a transcription factor involved in transcription of cytotoxicity-associated genes (Fig. 3b, upper panel). We then

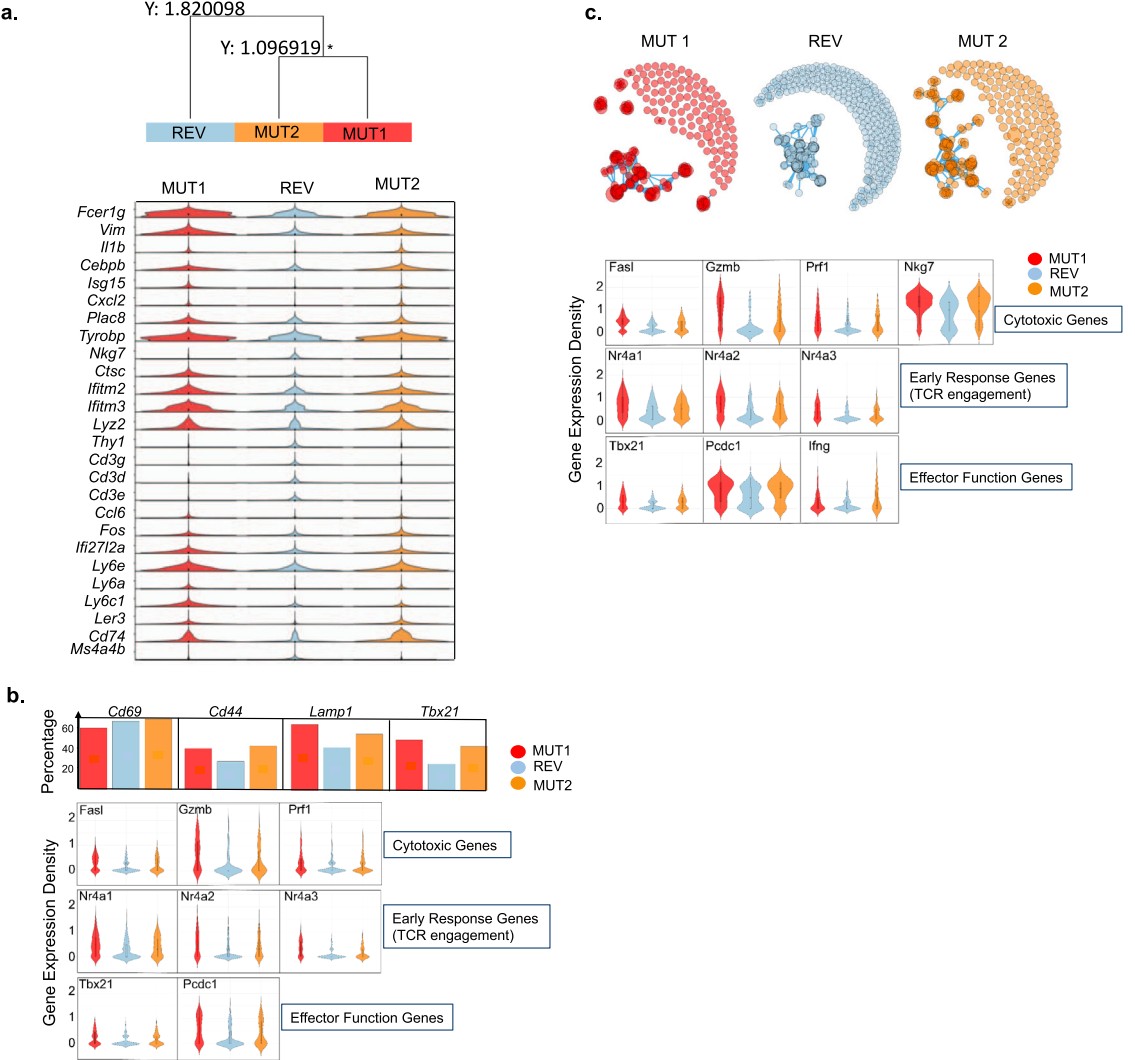

**Fig. 3 Intratumoral CD8 T cell response in mice challenged with the original and CRISPR edited Meth A cell lines.** Tumors from mice challenged with MUT1, REV, and MUT2 Meth A cell lines were harvested ten days after the tumor challenge. Tumor infiltrating CD45 + cells were sorted (Supplementary Fig. 9b) and analyzed by scRNA sequencing, as described in Methods. Combined scRNA sequencing data from the three resulting libraries were analyzed. **a** Top panel shows a simple hierarchical clustering of MUT1, REV and MUT2 libraries, based on the normalized average expression vector of top ~1,500 informative genes (selected by the highest average TF-IDF score) where, the Y value reflects the distance between clusters. The bottom panel represents the violin heat map plots of the top average TF-IDF scoring gene expression for the three libraries (expression of 26 genes is shown). Significant difference in distance of the REV versus MUT1 and MUT2 in the hierarchy is indicated by asterisk (please refer to Sup Fig. 6 for more details). **b** Top panel depicts the expression percentage of the genes involved in CD8 T cell activation of CD8 T cells that are derived from MUT1, MUT2, and REV libraries. Bottom panel shows the violin plots for the expression of genes involved in cytotoxicity, early response and other effector functions of CD8 T cells that are derived from the three libraries. **c** Top panel illustrates clone networks resulting from applying GLIPH-algorithm to the mixed pool of T cells TCR sequencing data, using igraph R package. Each node, represented by a circle, is a TCR clone. The diameter of a node is representative of the number of cells with the same TCR. Existence of a link between two nodes indicates global or local similarity between the two nodes as defined by the GLIPH algorithm. Further, a dense cluster in the network, characterized by high number of connections within a cluster and a low number of connections to neighboring clusters, suggest higher similarity and hence higher specificity within the cluster. A large number of dense clusters might suggest higher diversity in the network. Bottom panel represents the violin plots for the expression of genes involved in cytotoxicity, early response and other effector functions of the top 10 clonally expanded CD8 T cells that are derived from each library.

compared the gene expression patterns of the total immune cell population in TILs of MUT1, REV, and MUT2. The gene expression patterns in TILs of MUT1 and MUT2 showed a higher similarity to each other than to the REV: a simple hierarchical clustering (using Euclidean distance, and complete linkage) of the MUT1, REV and MUT2 libraries represented by the normalized average expression vector of top informative genes (selected by highest average TF-IDF score, see Methods) showed MUT1 and MUT2 are closer to each other (distance 1.097) than to the REV (distance 1.820) (Fig. 3a upper panel) with 97% confidence in the hierarchy edge/branch from Rev on one side to MUT1 and MUT2 on the other (see full details in Supplementary Fig. 7). In Fig. 3a bottom panel, where genes with variability among the three libraries are juxtaposed, it is clear that the difference between REV and MUT1/MUT2 is more pronounced than the difference between MUT1 and MUT2.

To identify differences in CD8 T cells of the three libraries, the RNA sequencing data of the combined libraries were clustered based on the gene expression pattern of each cell type as described in Methods. Annotation of the clusters was informed

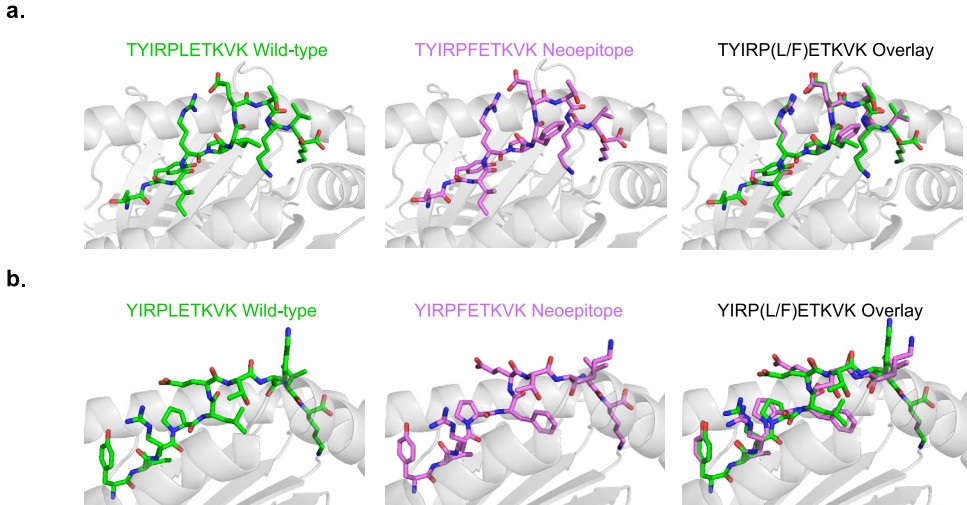

**Fig. 4 Models of peptide/MHC I complexes indicate structural and physical correlates with immunogenicity. a** For the tumor rejecting TYIRPFETKVK neoepitope, the leucine-to-phenylalanine substitution at position 6 is predicted to increase hydrophobic solvent accessibility by 17 Å², with the aromatic phenylalanine ring partially exposed for interactions with T cell receptors. An overlay of the neoepitope and its wild type counterpart demonstrates the substantial differences between the wild type peptide and neoepitope. **b** For the tumor rejecting YIRPFETKVK neoepitope, with the leucine-to-phenylalanine substitution now at position 5, the modeling predicts structural alterations in exposed side chains in response to the mutation, as well as a reduction in exposed hydrophobic solvent accessible surface area of 23 Å².

by both differentially expressed genes (DE Genes) and per cluster highly expressed genes identified by the TF-IDF analysis. T cells were re-clustered into 7 clusters by unsupervised clustering as described in Methods and enriched CD8 T cells populations were further analyzed (Supplementary Fig. 8a, b). Some cytotoxicity and effector function genes (*Tnf* (TNF), *Gzma* (Granzyme a) and *Ifng* (Interferonγ)) had similar expression pattern in CD8 T cells of all three libraries; however, the normalized average gene expression of other cytotoxicity genes (*Fasl* (Fas ligand), *Gzmb* (Granzyme b) and *Prf1* (Perforin1)) as well as other effector function genes (*Pcdc1* (PD1) and *Tbx21* (Tbet)) were significantly higher in CD8 T cells derived from MUT1 and MUT2 libraries compared to the REV library (*P* value < 0.001). Similarly, CD8 T cells of MUT1 and MUT2 libraries had a significantly higher expression than the REV library, of early response genes which are involved in TCR engagement (*Nr4a1*, *Nr4a2,* and *Nr4a3*, *P* value < 0.001) (Fig. 3b, bottom panel).

**T cell receptors (TCRs) in the TILs of MUT1, REV, and MUT2 tumors**. T cell receptors (TCRs) in the TILs of the three libraries were characterized using Grouping of Lymphocyte Interactions by Paratope Hotspots (GLIPH) analysis that groups together the TCRs into specificity groups based on the global and local similarities of the CDR3 regions of the TCRs[19]. Based on the GLIPH algorithm, 40–42.9% of all distinct clonotypes contributed to forming a network/similarity-based specificity groups in each of the libraries, while the rest were standalone clonotypes (with no similarity to other clonotypes). The similar percentage of network-based specificity groups (40–42.9%) in all three libraries was expected because of the existence of other mutations (except Ccdc85c^MUT) in all the three libraries. To further analyze the networks, we performed Louvain graph-based clustering of the specificity networks and calculated the modularity scores of the identified communities for each of the libraries (score of zero means the communities are the same and score of one refers to a perfect separation between communities). The modularity of a graph with respect to its division into communities measures how well separated (diverse) the different nodes (clonotypes) forming

the communities are from each other (see Methods). In the TILs, the TCR clonotypes that form communities/specificity groups are almost identical in frequency (42.9% for REV, and 42% and 40% for MUT1 and MUT2). However, the average modularity score of the communities/specificity groups including the most frequent (expanded) clonotypes is 0.53 for the REV, 0.73 for MUT1 and 0.77 for MUT2, indicating lower diversity of TCR clonotype in the TILs of REV than those of MUT1 and MUT2.

Using TCR-Seq analysis, top ten clonally expanded CD8 T cells were computationally pooled and further analyzed for gene expression patterns of their cytotoxic and effector functions. The normalized average gene expression of cytotoxicity-associated genes (*Fasl* or Fas ligand, *Gzmb* or Granzyme b, *Prf1* or Perforin1, *Nkg7* or Protein NKG7) as well as other effector function genes (*Tbx21* or Tbet, *Pcdc1* or PD1 and *Ifng* or Interferonγ) were significantly higher in the clonally expanded CD8 T cells derived from MUT1 and MUT2 libraries compared to the REV library (*Fasl P* value < 0.001, *Gzmb P* value < 0.001, *Prf1 P* value < 0.001 *Nkg7 P* value < 0.001, *Tbx21 P* value < 0.001, *Pcdc1 P* value < 0.001 and *Ifng P* value < 0.001). Similarly, the top 10 clonally expanded CD8 T cells of MUT1 and MUT2 libraries had a significantly higher expression of early response genes which are involved in TCR engagement (*Nr4a1* or NUR/77 *P* value < 0.001, *Nr4a2* or NUR-related factor 1 *P* value < 0.001 and *Nr4a3* or Orphan nuclear receptor TEC *P* value < 0.001) (Fig. 3c). Interestingly, *Ifng* and *Nkg7* which had a similar expression pattern in the pooled CD8 T cells of all three libraries (Supplementary Fig. 8c), had significantly higher expression in the top 10 clonally expanded CD8 T cells derived from MUT1 and MUT2 libraries compared to the REV library.

**Molecular modeling of Ccdc85c^MUT**. To gain insight into how the leucine to phenylalanine mutation leads to immunogenic epitopes, we modeled the structures of the 11-mer TYIRP-FETKVK neoepitope and the 10-mer YIRPFETKVK neoepitope bound to K^d. We modeled each corresponding WT peptide as well, to assess possible changes resulting from the mutation and thus infer how the neoepitopes might differ from self. We used

the same stochastic, template-based modeling procedure previously applied to murine neoepitopes[11]. For the TYIRPFETKVK 11-mer, the phenylalanine at position 6 is predicted to extend up from peptide near the MHC I α2 helix, increasing the amount of exposed hydrophobic surface 5% over the wild type peptide and potentially allowing the aromatic phenylalanine to interact with T cell receptors (Fig. 4a). Other than the side chain replacement, no conformational changes are predicted to occur in the peptide. For the YIRPFETKVK 10-mer, the new phenylalanine at position 5 is predicted to pack between the peptide and the a2 helix, in this case reducing exposed hydrophobic surface area (Fig. 4b). Subtle structural changes are predicted for the exposed side chains at positions 6 and 9, which could be suggestive of changes not captured by static structural modeling, such as changes in peptide flexibility that lead to altered TCR recognition[10].

## Discussion

A high binding affinity of peptides to MHC I is generally considered essential for immunogenicity[2,3]. However, some reports with cancer neoepitopes show that even peptides with very low affinities for MHC I elicit CD8-dependent tumor rejection[10,11]. These reports have used immunization with peptides to demonstrate immunogenicity. Here, we have asked and addressed if low affinity neoepitopes actually influence the natural immunogenicity of a tumor in vivo in the absence of artificial immunization. The answer is a clear affirmative. Using CRISPR to edit the cancer genome, our results show that introduction of a single point mutation into the Meth A tumor results in strong transcriptomic signatures of TCR engagement and cytotoxic functions in the CD8 T cells infiltrating the tumor. Extinction of this mutation eliminates that signature. Remarkably, the Ccdc85c[MUT] neoepitopes used here have very low affinities (IC50 values of 1,434 and 39,661 nM) for K[d]. These results have been obtained during examination of the natural growth of a tumor in the absence of any immunization and indicate that the low affinity MHC I-binding neoepitopes have a functional role in the immunogenicity of a tumor in vivo.

These findings are the most detailed yet, on the activity of a neoepitope that would be considered a non-MHC I binding epitope. Under the canonical view of MHC I-peptide interaction, epitopes with such low affinities are typically considered to be non-immunogenic and are routinely eliminated from further study. Our results show that such non-canonical neoepitopes indeed behave in manner similar to the traditional high affinity MHC I-binding epitopes, and in ignoring them, we run the risk of ignoring a significant proportion of the cancer immunome. Studies with several thousand cancer patients with a wide array of cancers have also noted the strong correlation between the presence of low affinity neoepitopes and good clinical outcomes[12,13,20].

## Methods

**Purification of MHC I eluted peptides**. MHC-I peptides were immunoaffinity purified as described before[11]. MethA cells or BMDCs were lysed and MHC-I molecules were immuno-affinity purified from cleared lysates with HIB antibodies cross-linked to Protein A-Sepharose 4B beads at 4 °C. MHC-I complexes and the bound peptides were eluted with 1% trifluoroacetic acid (TFA). Elutions containing MHC-I molecules were loaded in pre-conditioned Sep-Pak tC$_{18}$ 96-well plates (Waters). MHC-I peptides were eluted with 28% ACN in 0.1% TFA. Recovered peptides were dried using vacuum centrifugation (Thermo Fisher Scientific) and stored at −20 °C.

**LC-MS/MS analyses for the discovery of neoepitopes**. The LC-MS system consists of an Easy-nLC 1200 (Thermo Scientific, Bremen, Germany) coupled on-line to Q Exactive HF and or HF-X mass spectrometer (Thermo Scientific, Bremen, Germany). The LC-MS/MS parameters used for detection of TYIRPFETKVK and YIRPFETKVK peptides were previously reported[11] and here the parameter used for the detection of the peptide LYLDSKSDTTV are reported. The analytical

separation of the peptides was performed on a 500 mm homemade column of 75 μm inner diameter packed with ReproSil Pur C$_{18}$-AQ 1.9 μm resin (Dr. Maisch GmbH, Ammerbuch Entringen, Germany) during 120 min using a gradient of H$_2$O/FA 99.9%/0.1% and ACN/FA 95%/0.1%. For discovery MS spectra were acquired in the Orbitrap from $m/z = $ 300–1650 at a resolution of 60,000 ($m/z = $ 200) with a maximum injection time of 20 ms. The auto gain control (AGC) target value was set to 3e6 ions. MS/MS spectra were acquired at a resolution of 15'000 ($m/z = $ 200) using a 'top 10' data-dependent acquisition method. Each precursor ion was sequentially isolated with an isolation window of 1.2 $m/z$, activated by higher-energy collision dissociation (HCD) with a normalized collision-energy (NCE) of 27. Ions were accumulated to an AGC target value of 1e5 with a maximum injection time of 120 ms. In the case of assigned precursor ion charge-state of 4 and above, no fragmentation was performed. Selected ions were dynamically excluded for additional fragmentation for 20 s and the peptide match option was disabled.

**Identification of peptides by MS**. We employed the MaxQuant platform[21] version 1.5.5.1 to search the peak lists against a fasta file containing the mouse proteome (Mus musculus_UP000000589_10090, the reviewed part of UniProt, with no isoforms, including 24,907 entries downloaded in June 2016) concatenated to a list of 3,783 long peptides (up to 31 aa) encompassing the non-synonymous somatic mutations described above. The second peptide identification option in Andromeda was enabled. The enzyme specificity was set as unspecific. An FDR of 1% was required for peptides and no protein FDR was set. Peptides with a length between 8 and 25 amino acids were allowed. The initial allowed mass deviation of the precursor ion was set to 6 ppm and the maximum fragment mass deviation was set to 20 ppm. Methionine oxidation and N-terminal acetylation were set as variable modifications.

**Validation of neoepitopes with parallel reaction monitoring (PRM)**. Synthetic peptides labeled with heavy isotopes were purchased as crude (PEPotec SRM Custom peptide libraries grade 3) from ThermoFisher Scientific (Paisley, PA49RE, UK). For quality control, before spiking the peptides, we confirmed the absence of residual interferences of 'light' peptides by measuring separately each peptides with the method described below. The peptides were spiked into each of the peptidomic samples with a concentration of 100 fmol/μl. The PRM parameters used for detection of TYIRPFETKVK and YIRPFETKVK peptides were previously reported[11] and here the parameter for the detection of the peptide LYLDSKSDTTV are reported. The mass spectrometer was operated at a resolution of 120,000 (at $m/z = $ 200) for full scan MS, scanning a mass range from 300 to 1,650 $m/z$ with a maximum ion injection time of 120 ms and an AGC target value of 3e6. Then each peptide was isolated with an isolation window of 1.2 $m/z$ prior to ion activation by HCD (NCE = 27). Targeted MS/MS spectra were acquired at a resolution of 60,000 (at $m/z = $ 200) with a maximum ion injection time of 180 ms and an AGC target value of 1e6. The data were processed and analyzed by Skyline (MacCoss Lab, Skyline v19.1.0.193, Seattle, USA). An ion mass tolerance of 0.055 m/z was used to extract fragment ion chromatogram. Peptides with precursor's charge state z ≤ 3+ and fragment ion with z ≤ 2+ were used to monitor multiple transitions corresponding to –b and –y ion types. We plotted –y ion type transitions with z = 1 + . We then enabled synchronization of isotope labels for a proper alignment of transitions between heavy and endogenous peptides. Raw data were converted into Mascot generic format (mgf) by MSConvert (Proteowizard, Palo Alto, CA 94304, USA) in order to extract matched peak lists for heavy peptide and light counterpart for visualization of the MS/MS spectra. The assessment of MS/MS matching was done by pLabel (Version 2.4.0.8, pFind studio, Sci. Ac., China) and Skyline.

For the data shown in Fig. 1d, targeted MS-based detection of TYIRPFETKVK and YIRPFETKVK among MHC I peptides eluted from BMDCs pulsed with the 18-mer Ccdc85c[MUT]. Heavy labeled synthetic peptides were spiked into the peptide samples; the labeled amino acid is marked with a bold character and the mutation is in red. Matched peak lists for the "heavy" and "light" ions were extracted and monitored, while only single charge y ions were plotted. First, the absence of "light" peptide and the presence of the "heavy" peptide were confirmed by Parallel Reaction Monitoring (PRM) as a quality control measure in the synthetic peptide samples (upper left and lower left, respectively). Then, the co-elution of the synthetic "heavy" and endogenous "light" fragment ions was measured by PRM in Ccdc85c[MUT] pulsed BMDC MHC-I peptides. Figures were edited to improve resolution and readability.

**MHC II-peptide binding analysis**. MHC II-peptide binding analysis was conducted by cell-surface density assay[18] with modifications. NIH3T3 cells that stably express $H2$-$Aa1^d$ or -$Ea1^d$ were established through retrovirus-mediated transduction of cells with pMXs-puro[22] containing the full-length $H2$-$Aa1^d$ or -$Ea1^d$ with a C-terminal Strep-tag II (IBA GmbH), using packaging cell PLAT-E[22,23]. The stable cells were obtained by selection with puromycin (5 μg/ml) for two weeks. The expression constructs for the β subunit was designed to contain the signal peptide for HLA-DQB1*06:02, followed by the peptide via linkers[18], and the mature region of the β subunit (H2-Ab1$^d$ or H2-Eb1$^d$), with a C-terminal 6× His-tag. The construct was inserted into pMXs-IG that contains IRES and GFP

downstream of H2-Ab1[d] or H2-Eb1[d]. The stable H2-Aa1[d] or H2-Ea1[d] cells were transduced with retrovirus particles containing H2-Ab1[d]-peptide/pMXs-IG or IA-Eb1[d]-peptide/pMXs-IG. Cell-surface expression of H2-A[d] or H2-E[d] and cytoplasmic expression of GFP in GFP[+] MHC[+] cells were measured by flow cytometry 48 h after the transduction, using the following antibodies: H2-A[d], anti-mouse I-A[d] mAb with the dilution of 1:10, 20 µl per sample (39-10-8, BioLegend) or mouse IgG3 isotype control with the dilution of 1:100, 10 µl per sample (m078-3, clone 6A3, Medical & Biological Laboratories Co. Ltd.) and goat anti-mouse IgG3-PE with the dilution of 1:20, 15 µl per sample (sc-3767, Santa Cruz Biotechnology, Inc.); H2-E, anti-I-E[d] mAb with the dilution of 1:40, 10 µl per sample (115002, 14-4-4 S, Thermo Fisher) or mouse IgG2a isotype control with the dilution of 1:10, 10 µl per sample (m076-3, clone 6H3, Medical & Biological Laboratories Co. Ltd.) and goat F(ab')2 Anti-mouse Ig-PE with the dilution of 1:20, 20 µl per sample (1012-09, Southern Biotechnology Associates Inc.). The ratio of MHC MFI to GFP MFI (MHC/GFP) for each MHC II-peptide combination was calculated and normalized to the MHC/GFP for respective MHC II allele-G9 peptide. On each assay date, MHC/GFP for each MHC II-peptide combination was measured for three or four wells and their average was determined. The assay was repeated twice. Data were collected with SA3800 (Sony Imaging Products & Solutions Inc.) and analyzed using FCS Express 6 software (6.06.0022, De Novo Software, CA). The double-stranded DNA oligonucleotides encoding the signal sequence and peptide were synthesized (Genewiz Japan). The NIH3T3 cell line was obtained from the RIKEN Bioresource Center.

**Mice and tumors.** BALB/cJ mice (6–8 week-old females, stock # 000651) were purchased from the Jackson Laboratory and maintained in our specific pathogen-free mouse facilities under ethical approval from the Institutional Animal Care and Use Committee of the University of Connecticut School of Medicine. Experimental and control mice were kept in separate cages. Twelve light/12 dark cycle was used for mice housing. The temperature of the mice room and cages were kept around 65–75 °F. The humidity of the housing was 40–60%. Mice were euthanized by inhalation of carbon dioxide. Meth A cells that have been in our lab since 1988 were originally obtained from Lloyd J Old. Meth A cells were passaged in ascites and were determined to be free from mycoplasma contamination.

**Analysis of tumor growth.** Area under the curve (AUC) is used as a tool to measure tumor growth[24]. Briefly, AUC was calculated by selecting "Curves & Regression" and then "Area under curve" from the "analyze" tool, using the Prism 5.0 (GraphPad). A tumor rejection score (TRS) has been utilized for reporting the proportion of mice which reject tumors completely or near completely in response to vaccination with a given peptide. A maximum TRS of 5 indicates tumor rejection in 100% of the mice, and a TRS score of 0 indicates tumor rejection in no mice. The values between 0 and 5 are allocated based on the proportion of mice rejecting a tumor.

**Immunization.** Fifty microliter of TiterMax (CytRx Corporation) or Day 7 granulocyte-macrophage colony-stimulating factor-derived BMDCs (GM-CSF-BMDCs), were pulsed with 40 µg of the neoepitope. The peptide-pulsed BMDCs were used to immunize a single mouse. Immunizations were done twice, one week apart, and the mice were challenged with live tumor cells one week after the last immunization (Fig. 1b upper panel). All immunizations were performed in the presence of CTLA4 blockade, using the IgG2b isotype (Clone: 9D9, Bio X Cell), administered with the second immunization and every 3 days after tumor challenge. Peptides were synthesized by JPT Peptide Technologies.

**Bone marrow derived dendritic cells (BMDCs).** Bone marrow cells (2–3 million/10 cm$^2$ bacteriological Petri dishes) of 6- to 8-week-old mice were cultured in complete RPMI supplemented with 20 ng/ml recombinant murine GM-CSF (Peprotech) and incubated at 37 °C for 7 days to generate GM-CSF-BMDCs.

**Fluorescence activated cell sorting.** The antibody specific for Fixable Viability Dye eFluor® 780 (65-0865-14, dilution of 1:1000/sample) and CD45-PE (103106, clone: 30-F11, dilution of 1:50/sample) were purchased from eBioscience and Biolegend, respectively. Mouse FCR blocking reagent (130-092-575, dilution of 1:10/sample) was purchased from Miltenyi Biotec. Cell sorting was accomplished with FACS Aria II-B (Supplementary Fig. 9b).

**Single cell library generation.** Twelve thousand tumor infiltrating immune cells were loaded for capture, using a Chromium Single Cell 3′ Reagent Kits v2 Chemistry (10× Genomics)[25]. Following capture and lysis, complementary DNA was synthesized and amplified (12 cycles) as per the manufacturer's protocol. The amplified cDNA was used to construct Illumina sequencing libraries and was sequenced on a HiSeq4000 system (Illumina). The Cell Ranger Single-Cell Software Suite v.3.0 (10× Genomics) was used to perform sample demultiplexing, barcode processing and single-cell 3′ counting.

**Single cell RNA sequencing alignment, barcode assignment, and UMI counting.** Cell Ranger v.3.0 count pipeline was used to process the FASTQ files for

each sample. The mm10 genome and transcriptome was used to align samples, filter, and quantify. The "cellranger aggr" pipeline was used to aggregate the analysis files for each sample into a combined set by performing between-sample normalization (samples are subsampled for an equal number of confidently mapped reads per cell). Cell Ranger pipeline output, the 'feature (gene) vs cell' count matrix is then used for the secondary scRNA-Seq analysis in SC1 as described below[26].

**Single cell data analysis.** Samples from the libraries were analyzed using the SC1 tool available at sc1.engr.uconn.edu. Pre-processing quality control was conducted to exclude outlier and low quality cells based on the data distribution, from 20422 cells from 10× pipeline 15925 cells met our QC criteria (cells with over 30000 total UMIs or expressing less than 500 genes or over 5000 genes, with higher than 10% mitochondrial genes or less than 5% ribosomal genes were excluded from the analysis).

Cells were then clustered using Ward's Hierarchical Agglomerative Clustering algorithm using the top average TF-IDF genes as features[26], after log2($x$ + 1) transformation of the data. Clusters were annotated based on one-versus-all differential expression analysis between clusters, determined by a $p < 0.01$ and absolute value of $Log_2$ fold change of >1.

**Lymphoid population.** The Lymphoid compartment was re-clustered into six subclusters. Based on the differentially expressed genes, different subclusters were annotated as follows: CD8 T cells (assigned by their expression of *Cd3e* and *Cd8b1* and *Cd8a*), CD4 T cells (assigned by their expression of *CD3e* and *Cd8b*), Naive/early activated CD4 T cells (assigned by a high expression of *Sell*, *Il7r*, and *Ccr7* genes and lack expression of *Il2ra*, effector and cytotoxicity genes), proliferating CD4/CD8 T cells (defined by higher expression of *Stmn1* and *Mki67* genes and cell cycle gene expression analysis), NKC (defined by a high expression of *Ncr1*, *Nkg7*, and *Fcer1g*).

**Myeloid population.** The myeloid compartment was re-clustered into nine subclusters. Using DE gene list in myeloid compartment, different subclusters were annotated as follows: four clusters of macrophages (assigned by their expression of *Mrc1*, *Adgre1*, and *Itgam*), two clusters of DCs (assigned by their expression of *Zbtb46*, *Flt3*, and *H2-Oa*) two clusters of monocyte (assigned by their expression of Itgam, Ly6c, Ms4a4c and Il1b) and a neutrophil cluster (assigned by expression of *S100a8*, *S100a9*, and *Itgam*). Furthermore, for the cell types with more than one cluster (macrophages, DCs and monocytes), DE gene lists were generated to determine the main differences between different clusters of one cell type.

**Cell cycle gene expression analysis.** Using the Sc1 tool, each cluster was examined against the cell cycle gene list, obtained from GoTerm "Go:0007049". Clusters that were found to have a high expression of cell cycle genes and dominated by cell cycle effect were excluded from further analysis.

**TCR sequencing analysis.** Specificity groups/clusters in the TCR repertoire were identified via computational analysis following the grouping of lymphocyte interactions by paratope hotspots (GLIPH) algorithm from Glanville et al.[19]. GLIPH searches for global and local motif CDR3 similarity in TCR CDR regions with high contact probability. Each specificity group is analyzed in GLIPH for enrichment (of common V-genes, CDR3 lengths, clonal expansions, motif significance, and cluster size). Global similarity measures CDR3 differing by up to one amino acid and local similarity measures the shared enriched CDR3 amino acid motifs with 10× fold-enrichment and probability < 0.001. Supplementary Tables 2–4 show the enriched CDR3 motifs of TCRs from TILs of MUT1, REV and MUT2 libraries.

**Modularity score (as defined in igraph R package).** The modularity of a graph with respect to some division (or vertex types) measures how good the division is, or how separated are the different vertex types from each other. It defined as
$$Q = 1/(2\,m) * sum((Aij - ki * kj/(2\,m)) \, delta(ci, cj), i, j),$$
here m is the number of edges, $Aij$ is the element of the A adjacency matrix in row $i$ and

column $j$, $ki$ is the degree of $i$, $kj$ is the degree of $j$, $ci$ is the type (or component) of $i$, $cj$ that of $j$,

the sum goes over all i and j pairs of vertices, and delta($x$,$y$) is 1 if $x = y$ and 0 otherwise.

**Clustered regularly interspaced short palindromic repeats (CRISPR).** A guide RNA was designed that included the C - > A *Ccdc85c* mutation in its seed region (Supplementary Table 5). The seed region refers to the 8–12 nucleotides proximal to the PAM; mutations in this region significantly limit Cas9's ability to cleave target DNA. A single stranded oligodeoxynucleotide (donor ssODN) template was designed that contained 50-base pairs of homology to the endogenous sequence on either side of the target base (Supplementary Table 5). The donor ssODN was resuspended in TE buffer and stored according to the manufacturer recommendation upon receipt (IDT). Custom TrueGuide sgRNAs (Synthego) were resuspended to a concentration of 4 µg/µl in TE, aliquoted and stored at −20 °C. Prior to

transfection, Cas9 ribonucleoprotein complexes (Cas9 RNP) were formed by incubating 10 μg of Alt-R HiFi Cas9 Nuclease v3 (IDT) with 8 μg of sgRNA in 0.3 M NaCl for 30 min at room temperature. To produce revertant clones, Cas9 RNPs complexed as previously described and were mixed with 200 pmol of donor ssODN and delivered into $10^6$ MethA cells via electroporation with a Lonza 4D Nucleofector X Kit using program DS-150 and Cell Line Solution SG (Lonza) according to the manufacturer's protocols. Two days post-delivery, cells were split via limiting dilution into single-cell clones, allowed to expand for 21 days, and genotyped with PCR and Sanger sequencing at the *Ccdc85c* mutation locus in a 96-well plate. Clones that were successfully edited by CRIPR, were identified by Sanger sequencing.

**Structural modeling of wild type/neoepitope peptide-MHC pairs**. Structural modeling of the 9-mer, 10-mer, and 11-mer wild type and neoepitope peptide/MHC pairs was conducted as previously described[11]. Briefly, modeling utilized Rosetta[27,28] and the ref2015 energy function[29]. The structures used as templates for modeling were PDB 5T7G[30] for the 9-mer peptides and 5GSV[31] for the 10-mer and 11-mer peptides. The templates were energy minimized via Rosetta FastRelax[32]. As there was no 11-mer peptide/MHC structure containing H-2K$^d$ as of November 2019, the 11-mer was approximated by interpolating a glycine between residues 5 and 6 of the template. Subsequently, the desired peptide sequence was introduced via mutation of the template peptide. Structures were first modeled with a low resolution centroid kinematic closure protocol[33] then with a high resolution atomistic protocol. To sufficiently sample the available conformational space, we modeled 10,000 decoys for each peptide/MHC. The lowest scoring decoy of each was retained for further analysis. Root-mean-square deviation of atomic positions (RMSD) of peptide common or backbone heavy atoms between wild type and mutant peptides was calculated and models were inspected visually for differences in structural features. Solvent-accessible surface areas were calculated in Rosetta using a probe radius of 1.4 Å.

**Statistical analysis**. *P* values for comparisons of MHC/GFP MFI and AUC scores were calculated using *t*-test and 1-way ANOVA test, respectively, adjusted for multiple comparisons. *P* values were adjusted for multiple comparison by False Discovery Rate method or "Dunnet's multiple comparison test" or "Tukey's multiple comparison test". $P < 0.05$ was considered statistically significant. Differential expression (DE) analysis is done by performing *t*-test to compare clusters/libraries. The *t*-test uses the Welch (or Satterthwaite) approximation with 0.95 confidence interval by calling the *t*-test available in R stats package. Results of the Log$_2$ fold change and the *P* value from the analysis are provided with 1.5-fold change cutoff and 0.05 for *P* value.

**Reporting summary**. Further information on research design is available in the Nature Research Reporting Summary linked to this article.

## Data availability

Single Cell RNA-Seq and TCR-Seq data generated in this study have been deposited in the GEO database under accession code of GSE171100 and in the Supplementary Information Data file. There are no restrictions on data availability. Source data are provided with this paper.

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

## Acknowledgements

This research was funded in part by the Neag Cancer Immunology Translational Program, Northeastern Utilities Chair in Experimental Oncology, and the Personalized Immunotherapy Core Interest Group of the Connecticut Institute for Clinical and Translational Science, grant R35GM118166 from the US National Institutes of Health, the Indiana Clinical and Translational Science Institute (NIH grant UL1TR002529), the Ludwig Institute for Cancer Research, the ISREC Foundation and the Biltema Foundation. We thank Dr. Evan R. Jellison of the University of Connecticut School of Medicine for help with FACS sorting. We are grateful to Drs. Alessandro Sette and John Sidney for their help in measuring peptides affinities.

## Author contributions

P.S. and H.E.-N. conceptualized and designed the research, analyzed the data and wrote the manuscript; H.E.-N., R.E., W.C., H.M. and A.H. did the experiments. I.I.M. guided the analysis of RNA-seq data. M.M. designed and developed the methods for data analysis of single cell RNA-Seq. M.M., S.S. and T.S. analyzed the RNA-seq data. B.B. and G.K. performed the modeling of the peptides and MHC I. M.B.-S., J.M., H.S.P. and G.C. performed immunopeptidomics and MS analyses and interpreted the MS data.

## Competing interests

The authors declare no competing interests.
