## [Peer Review File · Nature Communications]

Reversion analysis reveals the in vivo immunogenicity of a poorly MHC I-binding cancer neoepitopeREVIEWER COMMENTS

Reviewer #1 (Remarks to the Author):

General Summary:

The authors assess the impact of low binding affinity neoepitopes on the natural immunogenicity of tumor. Current MHC-I binding affinity algorithms are being used to predict CD8+ immunogenicity with the assumption that a 'high-binder' is more likely to bind to MHC-1 and elicit CD8+ T cell response and thus result in greater anti-tumor immunity. However, studies have now suggested that low affinity binding neoantigens also elicit immune responses and an anti-cancer response. Similarly, the human trials that have based vaccine design on MHC-I binding affinity have not resulted in robust CD8+ T cell responses that are essential for anti-tumor activity. The authors perform both vaccination experiments and CRISPR to show that both vaccination against and expression (and lack thereof) of low binding neoantigens significantly impacts the overall immunogenicity of the tumor microenvironment. The authors use CRISPR to reverse expression of a pre-specified low-affinity neoantigen to WT sequence which reverses the immunogenic phenotype. Conversely, when the neoantigen is re-introduced, the immunogenic transcriptomic signature is restored. They also perform structural modeling to show that the MHC-neoepitope complex of these low-binding epitopes have increased hydrophobicity consistent leading to TCR reactivity.

Major Points:

Overall, this study is timely and is of interest to the field. These data underscore the idea that low affinity binding neoantigens should still be evaluated for immunogenicity and do impact the overall composition of the TME. These data are also important to address optimal neoantigen-targeted vaccine studies. Although the studies support the hypothesis that low-binding neoantigens are a source of immunogenicity and impact the TME, further studies to evaluate mechanism should be performed. Additionally, evaluation and characterization of neoantigen-specific T cells should be performed and ideally functional experiments should be carried out to directly demonstrate cytotoxic function of the low-binding neoantigen T cells.

Specific Points:

1. In the set of experiments 1a-1b, DCs pulsed with WT sequence should also be used as an additional control. Cross-reactivity to WT sequence should be tested by pulsing DCs with WT sequence as well, although we assume that WT is non-immunogenic. This is later done in fig 2a but using a peptide vaccine approach (details should be provided as to the consistency of vaccine - is an adjuvant used?)
2. It does look like there is some minimal anti-tumor activity of WT sequence vaccine in Fig 2a. ELISPOT assays could be used to measure immunogenicity of both the mutated and WT in conjunction with the tumor response curves.
3. There is no reference to a figure for the depletion study mentioned on lines 157-168. Additionally, CD4 depletion may also be indicated since vaccination is with a long peptide which could contain both Class I and II epitopes.
4. Although the authors conduct single-cell analyses, they do not characterize (mutation specific) neoantigen-specific T cells. Do these specific neoantigen-specific T cells mediate tumor rejection? Direct proof of this is necessary to fully support their hypotheses.

Reviewer #2 (Remarks to the Author):

Summary: The authors examined the effects of a low affinity class I neopeptide on CD8 T cell responses in mice challenged with Meth A sarcoma cells. For a particular neoantigen and corresponding peptides confirmed by MS in BMDCs, they used CRISPR to study homozygous and WT lines derived from the original heterozygous tumor. Although the neoantigens had low affinity, they found it could induce a CD8 T cell response characterized by a greater diversity of TCR clonotype and higher overall expression of early response genes involved in TCR engagement.

Predicting which mutations in a tumor are likely to elicit an immune response remains an important question for immunotherapy design and clinical application. High affinity and high expression of mutations remain the most predictive criteria for identifying mutations that have potential to generate responses, but are suboptimal, and field continues to search for understanding of what determines whether a mutation will be immunogenic. This study illustrates that low affinity may not be sufficient to rule out a mutation's potential to be immunogenic. Although the study focuses only a single mutation (with a second high affinity mutation provided for contrast), the cases study is instructive and will be of interest to the broader community investigating immunotherapies and antigen presentation in general.

The manuscript is well and clearly written. There are some concerns that should be addressed to further aid interpretation of the reported findings. In particular, more attention should be given to the role of MHC II presentation in driving the observed immune responses.

Major Comments:

The title and some of the language in the manuscript are misleading. The authors are likely trying to express that low affinity binders that would otherwise be ignored (as if they were 'non-binding') can drive immune responses. They should revise their title and manuscript to avoid the term 'non-binding' as that seems to suggest a non-MHC related mechanism which is clearly not the case.

The authors should further investigate the role of CD4 responses. That the cell surface density of class II in complex with WT and mutant peptides does not differ suggests that the binding affinity is likely similar, however it would be helpful to also show the binding affinity in a similar way to what was done for MHC I. A quick analysis using NetMHCII v3.2 shows that the 3 peptides mentioned (TYIRPFETKVK, YIRPFETKVK and IRPFETKVK) appear to have some potential to bind to H-2-IAb and H-2-IEd. Interestingly, this does not seem to be the case for the high affinity Alm1mut peptide (See pasted image). Is it possible that class II binding is driving the anti-tumor immune response to ccdc85cMut? Perhaps rather than contrasting to Alm1mut LYLDKSDTTV, the authors could try to identify a peptide that has high class I affinity AND class II affinity characteristics? Could the role of CD4 be assessed by including antibodies to CD4 T cells during the priming phase? If class II affinity is equally or more important than class I affinity, this would be important information for vaccine designs. Notably, in Ott et al (Nature 2017), they found predominantly CD4 responses to a peptide vaccine designed around peptides designed to elicit CD8 responses. The manuscript from Alspach et al in 2019 (also in Nature) showed the requirement for both CD8 and CD4 responses for effective immunity, though it is not clear if the same peptide must elicit both.

Note: See attached version for figure.

Minor comments:

Class II MHC typically presents longer peptides than class I. Did the authors consider using longer (e.g. 15mer) peptides for class II cell surface density assays?

For completeness, it would be helpful to provide a list of mutations in the tumor, their DNA and RNA variant allelic fraction, and perhaps predicted class I and class II affinities for peptides overlapping those mutations, perhaps limiting to only peptides where at least one of the 2 has weak affinity. This would allow the ccdc85c mutation to be placed into context of the overall mutation landscape.

The authors state "At the same time, two retrospective human studies analyzing the genomic and clinical outcome data from nearly 7,000 patients with 27 cancer types, have shown that better clinical outcomes and T cell infiltration of tumors are associated with the presence of cancer neoepitopes with low affinities for HLA I molecules, and not with the presence of high affinity HLA I-binding neoepitopes" [12,13] This sentence does not fully represent the two papers. The first seems to focus only on part of the TCGA and fewer than 1000 patients treated with ICB. Perhaps the authors could be more explicit about the clinical outcomes and how many patients were

associated with each type. The second paper focuses on agretopicity (i.e. the relative mutant vs wildtype affinity) which they term ADNs or alternatively defined neopeptides, and finds these are predictive of survival/outcomes. The authors should more clearly point to the evidence in these papers that demonstrates that clinical outcomes are not associated with high affinity binders as they state "not with the presence of high affinity HLA-I binding neopeptides".

The authors may wish to clarify in the introduction that there are many reports where higher affinity neopeptides are immunogenic. The problem is that the majority of mutations are unlikely to generate immunogenic neopeptides and the use of affinity as a criteria for prioritizing neopeptides can help reduce false positives. However, as the authors demonstrate here, throwing out all mutations with low affinity may in some cases be "throwing the baby out with the bath water". This manuscript illustrates this with a compelling case study, though further analysis

Tumor rejection score (TRS) is mentioned in the first results paragraph to be described in the Methods but no section was found.

There is a typo in CRISPR methods section: "CRIPR" in last line

Reviewer #3 (Remarks to the Author):

The demonstration that MHC-I-restricted presentation of low affinity peptides represents a relevant source of rejection antigens is indeed an important contribution of these authors to the cancer immunology and immunotherapy fields. The leading investigators of the present manuscript recently published an article demonstrating this using the MethA mouse tumor model (Ebrahimi-Nik et al, JCI Insight 2019). The present study aims to deepen these findings by studying changes in the tumor microenvironment of MethA tumors carrying a point mutation in the Ccdc85c gene. Similar to what was shown in their previous work, immunization with an 18-aminoacid-length peptide containing the mutant peptide (L-to-F; Ccdc85cMUT) near the center induces strong tumor rejection. Authors aimed to map the MHC-class I-restricted epitope peptide by immunizing mice with overlapping truncated peptides and evaluating loss of anti-tumor protection. The authors concluded that the 10-mer YIRPFETKVK is the MHC-class I-restricted epitope responsible of anti-tumor protection. Based on MS analyses of MHC-I eluted peptides, the authors concluded that the 11-mer TYIRPFETKVK and 10-mer YIRPFETKVK neopeptides are actually presented by BMDCs pulsed with the 18-mer Ccdc85cMUT peptide. Then, the authors analyze the tumor microenvironment of mice immunized with Ccdc85cMUT or Alms1MUT, as a control, by scRNAseq observing that the former leads to higher lymphoid infiltration, including activated CD4 and CD8 T cells displaying higher levels of transcripts associated with cytotoxicity, effector function and TCR engagement. Then, authors analyzed the tumor microenvironment employing an experimental setting that does not involve immunization by comparing MethA tumors expressing the mutant (L-to-F; Ccdc85cMUT) gene in either one (MUT1), neither (REV) or both (MUT2) alleles. Similar to the immunization setting, they observed transcript expression skewed towards cytotoxic activity, effector function and TCR engagement in tumors carrying one (MUT1) or two (MUT2) mutant alleles as compared to REV. This differential transcriptomic state is also observed in clonally expanded T cells.

Even if some of the results presented in this manuscript are potentially interesting, a series of concerns have been raised by this reviewer.

Major comments:

The present study falls short to provide a mechanistic insight on why low MHC-I affinity neopeptides can function as rejection antigens.

The 11-mer epitope described here was already identified in their previous study (Ebrahimi-Nik et al, JCI Insight 2019), rendering some of the reported experiments overlapping. Actually, the contribution of CD8 T cells to anti-tumor protection in this immunization-challenge model, refers to anti-CD8 depletion experiments reported in their previous publication.

It is not clear to this reviewer the experimental setting of the immunization/challenge experiments (one or more immunizations how many days apart?, the interval between immunizations and tumor challenge, etc). A full description of the experimental setting is necessary. Also, a scheme of the experimental setting would be helpful.

The way to show tumor protection data, as Tumor rejection score (TRS), is not defined in methods' section or elsewhere, and represents a rather unconventional way to do so.

More importantly, grouping tumor growth curves from mice immunized with different truncated peptides does not allow to compare the protection elicited among the different truncated epitopes. It would be much more informative to display the individual tumor growth curves (and average perhaps) for the different truncated peptides to be able to understand how protective are the different peptides. In this regard, it is not clear why tumor protection diminishes from 5.0 to 3.0-3.3 (significant?) when mice are immunized with peptides STYIRPFETKVK, PSSTYIRPFETKVKLL and SSTYIRPFETKVKL, all which contain the predicted MHC-class-I-restricted epitopes.

To address the contribution of T cell populations to anti-tumor activity in the immunization setting, these experiments should include CD4 and CD8 depletion experiments. In their previous paper, the authors performed CD8 depletion experiments during the priming phase? It would be better to do the depletion right before the tumor challenge, in particular for CD4 T cells which may be important during the priming phase.

It seems quite odd that tumor growth data of mice immunized with the wild-type non-mutated Ccdc85cWT peptide (Fig. 2a) is displayed here. It seems more obvious to display these data at the beginning in Fig. 1b. Instead, this figure could start with the evidence showing that control immunization with Alms1MUT, does not lead to tumor rejection because this is the control used in the experiments described in Figure 2b-d. Figure 2c helps to define the different tumor-infiltrating cells, importantly clusters 1 and 5 which are further analyzed. How can immunization with another neoepitope such as Alms1MUT LYLDKSDTTV, which has a higher MHC-I affinity, not induce comparable responses

In the experimental setting that does not involve immunization, enhanced immunogenicity of the low MHC-I affinity neoepitope should be functionally demonstrated in terms of tumor growth. If the presence of the mutant neoepitope is relevant, tumor growth should be affected either spontaneously or after administration of checkpoint blockade immunotherapies such as anti PD1 and/or CTLA-4. Depletion experiments should also be performed to address the contribution of CD8 and CD4 T cell populations.

This manuscript would greatly improve by a robust analysis of Ccdc85cMUT-specific CD8 T cell responses. Authors should quantify and phenotypically characterize CD8 T cell responses specific for both wild-type (T)YIRPLETKVK- and mutated (T)YIRPFETKVK epitopes. To this end, they could perform flow cytometry-based analyses, such as MHC-multimer staining and/or intracellular cytokine staining after ex vivo peptide stimulation.

Along the different figures, statistical analysis is not informed for all relevant comparisons. The number of individual experiments performed for each analysis should be described.

Minor comments

The title does not quite reflect the main findings of the manuscript. It is not needed to remark a technical aspect of the study. In addition, to name neoepitopes with low affinity for MHC-class-I molecules as "non-MHC binders" seems misleading.

It is not possible to assure that the 18-mer is presented through cross-presentation by BMDCs. Alternatively, this peptide may potentially be degraded extracellularly generating shorter peptides that can be loaded onto MHC-I molecules, bypassing antigen cross-presentation. This concern is particularly latent, when using relatively high concentrations (100 μ M) of the peptide, as inferred by the authors' previous publication.

REVISED Point-by-point response

MS# NCOMMS-20-47737

Revised title: Reversion analysis reveals the immunogenicity in vivo of a poorly MHC I-binding cancer neoepitope

Ebrahimi-Nik et al.

Reviewer #1 (Remarks to the Author):

General Summary:

The authors assess the impact of low binding affinity neoepitopes on the natural immunogenicity of tumor. Current MHC-I binding affinity algorithms are being used to predict CD8+ immunogenicity with the assumption that a 'high-binder' is more likely to bind to MHC-1 and elicit CD8+ T cell response and thus result in greater anti-tumor immunity. However, studies have now suggested that low affinity binding neoantigens also elicit immune responses and an anti-cancer response. Similarly, the human trials that have based vaccine design on MHC-I binding affinity have not resulted in robust CD8+ T cell responses that are essential for anti-tumor activity. The authors perform both vaccination experiments and CRISPR to show that both vaccination against and expression (and lack thereof) of low binding neoantigens significantly impacts the overall immunogenicity of the tumor microenvironment. The authors use CRISPR to reverse expression of a pre-specified low-affinity neoantigen to WT sequence which reverses the immunogenic phenotype. Conversely, when the neoantigen is re-introduced, the immunogenic transcriptomic signature is restored. They also perform structural modeling to show that the MHC-neoepitope complex of these low-binding epitopes have increased hydrophobicity consistent leading to TCR reactivity.

Major Points:

Overall, this study is timely and is of interest to the field. These data underscore the idea that low affinity binding neoantigens should still be evaluated for immunogenicity and do impact the overall composition of the TME. These data are also important to address optimal neoantigen-targeted vaccine studies.

We are deeply thankful to the reviewers for appreciating the significance of our findings.

Although the studies support the hypothesis that low-binding neoantigens are a source of immunogenicity and impact the TME, further studies to evaluate mechanism should be performed. Additionally, evaluation and characterization of neoantigen-specific T cells should be performed and ideally functional experiments should be carried out to directly demonstrate cytotoxic function of the low-binding neoantigen T cells.

Please see the responses below.

Specific Points:

1. In the set of experiments 1a-1b, DCs pulsed with WT sequence should also be used as an additional control. Cross-reactivity to WT sequence should be tested by pulsing DCs with WT sequence as well, although we assume that WT is non-immunogenic. This is later done in fig 2a but using a peptide vaccine approach (details should be provided as to the consistency of vaccine - is an adjuvant used?)

We thank the reviewer for this comment and we now show the data with the WT sequences in Fig. 1b. lower panel. Actually, the WT data previously shown in Fig. 2 (but now moved to Fig.1) were

generated by using the DCs pulsed with the WT peptide, and not with another adjuvant, and this point is also clarified in the figure legend.

2. It does look like there is some minimal anti-tumor activity of WT sequence vaccine in Fig 2a. ELISPOT assays could be used to measure immunogenicity of both the mutated and WT in conjunction with the tumor response curves.

We understand why the reviewer feels that there might be a minimal anti-tumor activity in the WT sequence. In fact, there is not even minimal activity. Please note the P value. Also, this is only one of the several experiments we have carried out with the WT sequence; we have never seen even a modest reproducible activity in the WT sequence.

3. There is no reference to a figure for the depletion study mentioned on lines 157-168. Additionally, CD4 depletion may also be indicated since vaccination is with a long peptide which could contain both Class I and II epitopes.

We apologize. We have now added the following text to refer to the figure:

“The anti-tumor activity of *Ccdc85c^{MUT}* was abrogated by depleting the mice of CD8 cells by treating the mice with the anti-CD8 antibody but not by a control antibody during the priming phase as previously described (Ebrahimi-Nik et al., 2019, Fig. 3C).

We did consider the possibility that the 18-mer long peptide could contain both Class I and II epitopes, hence our analysis looking for a class II epitope in the three tumor-rejection-enabling peptides TYIRPFETKVK, YIRPFETKVK and IRPFETKVK and their WT counterparts; no structural basis of better binding by the neoepitope was detected as described in the text on page 8 and shown in Supplementary Fig. 1 and Supplementary Table 1.

We also attempted CD4 depletion as suggested by the reviewer. The problem is that using anti-CD4 antibodies to deplete CD4 T cells results not only in depletion of helper T cells (which may be helping the CD8 response) but also the regulatory T cells (which may be suppressing the CD8 response). Hence, CD4 depletion will not solve the problem. Indeed, our experiment gave us precisely the kind of ambiguous response reflective of the two-sides-of-the-knife activity of CD4 cells: CD4 depletion did have an influence on tumor growth in 2/5 mice but no influence whatsoever in the remaining 3/5 mice. We now show this observation in Supplementary Fig. 1b. This variability exists, in our thinking, because of variability of Treg activity in individual mice. Unfortunately, the reagents that can distinguish between the helper and the regulatory T cells do not exist at the moment. An indirect way to look for a contribution of CD4 cells was to look for the presence of MHC II neoepitopes within our active sequences. As described in the text on page 8 and shown in Supplementary Fig. 1 and Supplementary Table 1, we could not find such epitopes.

4. Although the authors conduct single-cell analyses, they do not characterize (mutation specific) neoantigen-specific T cells. Do these specific neoantigen-specific T cells mediate tumor rejection? Direct proof of this is necessary to fully support their hypotheses.

We fully appreciate the validity of this comment. That being said, the known methodologies for characterizing antigen-specific T cells are not available to us precisely because the low affinity of the peptides for MHC I makes it impossible to generate tetramers for them. We now say so on page 6. In order to strengthen our case, we have now added data (see figure below, also added in the manuscript as Supplementary Fig. 4b) that show that editing away of the neoepitope to the wildtype sequence results into a tumor that grows more rapidly and that difference is statistically significant.

Sup. Fig. 4. b. BALB/cJ mice were tumor challenged with either REV or MUT cell line. Each line represents tumor growth in a single mouse. On the right panel, total Area Under the Curve (AUC) scores for REV and MUT are plotted. Each bar shows the average total AUC score for the indicated group. Error bars represent standard deviation (SD). *P* values were calculated using t-test.

Reviewer #2 (Remarks to the Author):

Summary: The authors examined the effects of a low affinity class I neopeptide on CD8 T cell responses in mice challenged with Meth A sarcoma cells. For a particular neoantigen and corresponding peptides confirmed by MS in BMDCs, they used CRISPR to study homozygous and WT lines derived from the original heterozygous tumor. Although the neoantigens had low affinity, they found it could induce a CD8 T cell response characterized by a greater diversity of TCR clonotype and higher overall expression of early response genes involved in TCR engagement.

Predicting which mutations in a tumor are likely to elicit an immune response remains an important question for immunotherapy design and clinical application. High affinity and high expression of mutations remain the most predictive criteria for identifying mutations that have potential to generate responses, but are suboptimal, and field continues to search for understanding of what determines whether a mutation will be immunogenic. This study illustrates that low affinity may not be sufficient to rule out a mutation's potential to be immunogenic. Although the study focuses only a single mutation (with a second high affinity mutation provided for contrast), the cases study is instructive and will be of interest to the broader community investigating immunotherapies and antigen presentation in general.

The manuscript is well and clearly written. There are some concerns that should be addressed to further aid interpretation of the reported findings. In particular, more attention should be given to the role of MHC II presentation in driving the observed immune responses.

We are deeply thankful to the reviewers for appreciating the significance of our findings.

Major Comments:

The title and some of the language in the manuscript are misleading. The authors are likely trying to express that low affinity binders that would otherwise be ignored (as if they were 'non-binding') can drive immune responses. They should revise their title and manuscript to avoid the term 'non-binding' as that seems to suggest a non-MHC related mechanism which is clearly not the case.

We have changed the title to: "Reversion analysis reveals the immunogenicity in vivo of a poorly MHC I-binding cancer neoepitope"

The authors should further investigate the role of CD4 responses. That the cell surface density of class II in complex with WT and mutant peptides does not differ suggests that the binding affinity is likely similar, however it would be helpful to also show the binding affinity in a similar way to what was done for MHC I.

Please note that the measured value in cell-surface MHC density assay correlates well with the actual affinity of the peptide to MHC II, as now indicated on page 8.

A quick analysis using NetMHCII v3.2 shows that the 3 peptides mentioned (TYIRPFETKVK, YIRPFETKVK and IRPFETKVK) appear to have some potential to bind to H-2-IA^d and H-2-IE^d.

Potential binding to MHC II, in and of itself, is not relevant. What is relevant is the differential binding between the mutant and WT, as shown by us and by others in the mouse and human systems (Duan et al. 2014, Ebrahimi-Nik et al. 2019, Ghorani et al. 2018, Rech et al. 2019). Binding predictions by NetMHCII v3.2 (see Figure below – not included in the manuscript but shown here for the reviewer’s perusal only) indicate that wild type and mutant peptides do not show differential binding to MHC II. The Figure represents the binding affinities of all four possible 15-mer Ccdc85c peptides that can be extracted from 18-mer Ccdc85c peptide. No noticeable difference was observed between the wild type and their mutant counterparts. Although this fact has only been depicted for 15-mer peptides, it holds true for other peptide lengths (12 – 18-mer) as well. Moreover, the precise peptide which had full tumor rejection capacity (10 – 11-mer) was examined using the same tool, where it was found that the binding affinities were worse than 8,000nM for H2-IA^d and H2-IE^d.

Predicted binding affinity of Ccdc85c^{MUT} and Ccdc85c^{WT} to H2-IA^d and H2-IE^d. Using NetMHCII v3.2, binding affinity of wild type (black bars) and mutant (red bars) Ccdc85c peptides to H2-IA^d and H2-IE^d are predicted.

Interestingly, this does not seem to be the case for the high affinity Alm1mut peptide (See pasted image). Is it possible that class II binding is driving the anti-tumor immune response to ccdc85cMut?

As discussed above (see Figure above and accompanying text) and discussed below (in the answer to the question “Could the role of CD4 be assessed.....”), we were not able to see any case for an effective MHC II-binding neopeptide, nor a case for a role for CD4 cells in immunity elicited by Ccdc85c^{MUT} even though we looked for the evidence as we described partly in the original submission, and more fully, in this revised submission and in our answers here.

Perhaps rather than contrasting to Alm1mut LYLDKSDTTV, the authors could try to identify a peptide that has high class I affinity AND class II affinity characteristics?

The reviewer will hopefully agree that in light of our answer to the previous question, this issue becomes moot.

Could the role of CD4 be assessed by including antibodies to CD4 T cells during the priming phase? If class II affinity is equally or more important than class I affinity, this would be important information for vaccine designs.

We have indeed analyzed the role of CD4 responses. It is for reason that we tried to determine if any peptides within Cdc85c^{MUT} could be presented by MHC II molecules. As mentioned on page 8 (as well and figure above), we found no evidence of differential binding of mutant versus WT peptides to MHC II. We also attempted CD4 depletion as suggested by the reviewer. The problem is that using anti-CD4 antibodies to deplete CD4 T cells results not only in depletion of helper T cells (which may be helping the CD8 response) but also the regulatory T cells (which suppress the CD8 response). Hence, CD4 depletion will not solve the problem. Indeed, our experiment gave us precisely the kind of ambiguous response reflective of the two-sides-of-the-knife activity of CD4 cells: CD4 depletion did have an influence on tumor growth in 2/5 mice but no influence whatsoever in the remaining 3/5 mice. We now show this observation in Supplementary Fig. 1b. This variability exists, in our thinking, because of variability of Treg activity in individual mice. Unfortunately, the reagents that can distinguish between the helper and regulatory T cells do not yet exist. The lack of the presence of an MHC II neoepitope, as discussed in the previous response, also attests to a lack of role for a CD4 response.

Notwithstanding the above, the novelty of our manuscript resides not in CD8 or CD4 dependence or lack thereof, but in the fact that a neoepitope with a poor affinity for MHC I shows activity in vivo. The questions about CD8 or CD4-dependence are completely legitimate; however, they do not bear on the novelty of our studies.

Notably, in Ott et al (Nature 2017), they found predominantly CD4 responses to a peptide vaccine designed around peptides designed to elicit CD8 responses. The manuscript from Alspach et al in 2019 (also in Nature) showed the requirement for both CD8 and CD4 responses for effective immunity, though it is not clear if the same peptide must elicit both.

These are elegant studies but considering that they do not see any clinical responses that can be clearly attributed to the neoepitopes, the CD4 / CD8 responses reported in them, are irrelevant. Several studies show nice CD4 or CD8 responses but such responses have no clinical correlate. Such responses, in and of themselves, do not tell us much.

Minor comments:

Class II MHC typically presents longer peptides than class I. Did the authors consider using longer (e.g. 15mer) peptides for class II cell surface density assays?

We tested the three peptides which elicited complete or near complete tumor rejection since those sequences alone were necessary and sufficient for anti-tumor activity. If an MHC II neoepitope had to play a role, it had to be within those sequences.

For completeness, it would be helpful to provide a list of mutations in the tumor, their DNA and RNA variant allelic fraction, and perhaps predicted class I and class II affinities for peptides overlapping those mutations, perhaps limiting to only peptides where at least one of the 2 has weak affinity. This would allow the *ccdc85c* mutation to be placed into context of the overall mutation landscape.

We have now referred to this information published by us in Duan et al. 2014 in the Supplementary Information. We have added on page 6 to the text:

“(See Duan et al. Supplementary Table S1 for a list of all mutations and predicted neoepitopes of the Meth A sarcoma).”

The authors state “At the same time, two retrospective human studies analyzing the genomic and clinical outcome data from nearly 7,000 patients with 27 cancer types, have shown that better clinical outcomes and T cell infiltration of tumors are associated with the presence of cancer neoepitopes with low affinities for HLA I molecules, and not with the presence of high affinity HLA I-binding neoepitopes” [12,13] This sentence does not fully represent the two papers. The first seems to focus only on part of the TCGA and fewer than 1000 patients treated with ICB. Perhaps the authors could be more explicit about the clinical outcomes and how many patients were associated with each type. The second paper focuses on *agretopicity* (i.e. the relative mutant vs wildtype affinity) which they term ADNs or alternatively defined neopeptides, and finds these are predictive of survival/outcomes. The authors should more clearly point to the evidence in these papers that demonstrates that clinical outcomes are not associated with high affinity binders as they state “not with the presence of high affinity HLA-I binding neoepitopes”.

Please note that the term *Alternatively Defined Neoepitopes* refers to low-affinity neoepitopes which have a high *Differential Agretopicity* as defined by us. We have now added the following on page 4:

“As a specific example, Rech et al. (2018) show in Fig. 4 in their paper using the TCGA database that the presence of high-affinity neoepitopes, or *Conventionally Defined Neoepitopes* in lung and bladder cancers as well as melanomas did not correlate with survival, while the presence of low-affinity neoepitopes (or *Alternatively Defined Neoepitopes*), did. Similar results are obtained from analysis of an independent database of melanoma patients (Van Allen et al. 2015). Rech et al. (2018) conclude that “In TCGA bladder and lung squamous tumor types, a high top three score for ADNs, but not CDNs, also correlated strongly with improved survival (Fig. 4C). Overall, significantly improved survival was observed in high ADN, but not CDN, load samples in multiple TCGA tumor types after adjustment for multiple hypothesis testing...”.”

The authors may wish to clarify in the introduction that there are many reports where higher affinity neoepitopes are immunogenic. The problem is that the majority of mutations are unlikely to generate immunogenic neoepitopes and the use of affinity as a criteria for prioritizing neopeptides can help reduce false positives. However, as the authors demonstrate here, throwing out all mutations with low affinity may in some cases be “throwing the baby out with the bath water”. This manuscript illustrates this with a compelling case study, though further analysis

We thank the reviewer for recognizing the contribution of our data. We already have pointed out, as the reviewer asks us to do, that "there are many reports where higher affinity neopeptides are immunogenic". Please note that we already say in the Introduction that "a number of high affinity neopeptides that can elicit tumor rejection as well as CD8 T cell responses measurable in vitro, have been identified (Castle et al., 2012; Gubin et al., 2014; Yadav et al., 2014)". In fact, these three papers represent perhaps all the studies where high affinity neopeptides have been shown to mediate tumor rejection.

Tumor rejection score (TRS) is mentioned in the first results paragraph to be described in the Methods but no section was found.

We apologize; this has now been corrected and the Methods now contains the information.

There is a typo in CRISPR methods section: "CRIPR" in last line

Thank you. This has been corrected.

Reviewer #3 (Remarks to the Author):

The demonstration that MHC-I-restricted presentation of low affinity peptides represents a relevant source of rejection antigens is indeed an important contribution of these authors to the cancer immunology and immunotherapy fields. The leading investigators of the present manuscript recently published an article demonstrating this using the MethA mouse tumor model (Ebrahimi-Nik et al, JCI Insight 2019). The present study aims to deepen these findings by studying changes in the tumor microenvironment of MethA tumors carrying a point mutation in the Ccdc85c gene. Similar to what was shown in their previous work, immunization with an 18-aminoacid-length peptide containing the mutant peptide (L-to-F; Ccdc85cMUT) near the center induces strong tumor rejection. Authors aimed to map the MHC-class I-restricted epitope peptide by immunizing mice with overlapping truncated peptides and evaluating loss of anti-tumor protection. The authors concluded that the 10-mer YIRPFETKVK is the MHC-class I-restricted epitope responsible of anti-tumor protection. Based on MS analyses of MHC-I eluted peptides, the authors concluded that the 11-mer TYIRPFETKVK and 10-mer YIRPFETKVK neoepitopes are actually presented by BMDCs pulsed with the 18-mer Ccdc85cMUT peptide. Then, the authors analyze the tumor microenvironment of mice immunized with Ccdc85cMUT or Alms1MUT, as a control, by scRNAseq observing that the former leads to higher lymphoid infiltration, including activated CD4 and CD8 T cells displaying higher levels of transcripts associated with cytotoxicity, effector function and TCR engagement. Then, authors analyzed the tumor microenvironment employing an experimental setting that does not involve immunization by comparing MethA tumors expressing the mutant (L-to-F; Ccdc85cMUT) gene in either one (MUT1), neither (REV) or both (MUT2) alleles. Similar to the immunization setting, they observed transcript expression skewed towards cytotoxic activity, effector function and TCR engagement in tumors carrying one (MUT1) or two (MUT2) mutant alleles as compared to REV. This differential transcriptomic state is also observed in clonally expanded T cells.

Even if some of the results presented in this manuscript are potentially interesting, a series of concerns have been raised by this reviewer.

Major comments:

The present study falls short to provide a mechanistic insight on why low MHC-I affinity neoepitopes can function as rejection antigens.

The 11-mer epitope described here was already identified in their previous study (Ebrahimi-Nik et al, JCI Insight 2019), rendering some of the reported experiments overlapping. Actually, the contribution of CD8 T cells to anti-tumor protection in this immunization-challenge model, refers to anti-CD8 depletion experiments reported in their previous publication.

Certainly, the reviewer would agree that the same model system can be used in many distinct studies. The epitope was used in our previous study, and its CD8-dependence was demonstrated in that study, as we clearly point out in this manuscript in its original version. There is NO OVERLAP between experiments, nor between the points made between the two papers. Our

previous paper did nothing to test the immunogenicity in vivo of a naturally growing tumor which does or does not have this low-affinity neopeptide.

It is not clear to this reviewer the experimental setting of the immunization/challenge experiments (one or more immunizations how many days apart?, the interval between immunizations and tumor challenge, etc). A full description of the experimental setting is necessary. Also, a scheme of the experimental setting would be helpful.

We apologize. The details have now been added in Methods under "Immunization". A schema of the immunization-challenge experiment is shown in Fig. 1b upper panel.

The way to show tumor protection data, as Tumor rejection score (TRS), is not defined in methods' section or elsewhere, and represents a rather unconventional way to do so.

We apologize for the omission and we now do so on page 21. This method is un-conventional only because such large analysis of the tumor rejection eliciting capacity of over 20 individual peptides involving hundreds of mice has not been published before. We had to find a relatively straight forward method to convey a vast amount of information (see Figure in response to the next question), and this seemed to be a straight forward way.

More importantly, grouping tumor growth curves from mice immunized with different truncated peptides does not allow to compare the protection elicited among the different truncated epitopes. It would be much more informative to display the individual tumor growth curves (and average perhaps) for the different truncated peptides to be able to understand how protective are the different peptides.

As mentioned above, these are vast amounts of data. We considered various way to show them and decided on the format that we have used, because we did not see what additional information will be evident by showing the individual data. If the reviewer has a different format in mind, we are happy to comply. Please see below for the reviewer's perusal; we are not formally including it in the submission but are happy to do so if so instructed.

Several truncated versions of the 18-mer *Ccdc85c^{MUT}* peptide were tested in tumor rejection assay. BALB/c mice were immunized and tumor challenged. Each line represents tumor growth in a single mouse. Tumor rejection score (TRS) for each group of neopeptides is shown, where five represents a complete tumor protection and zero means no tumor rejection.

In this regard, it is not clear why tumor protection diminishes from 5.0 to 3.0-3.3 (significant?) when mice are immunized with peptides STYIRPFETKVK, PSSTYIRPFETKVKLL and SSTYIRPFETKVKL, all which contain the predicted MHC-class-I-restricted epitopes.

We appreciate the reviewer's point; however, the result is not as surprising as it seems. A number of studies (as just one example, see Ma et al. 2009 *Journal of Immunology*, *The amino acid sequences flanking an antigenic determinant can strongly affect MHC class I cross-presentation without altering direct presentation.*) have pointed out the powerful influence of flanking sequences on the ability of long peptides to be cross-presented.

To address the contribution of T cell populations to anti-tumor activity in the immunization setting, these experiments should include CD4 and CD8 depletion experiments. In their previous paper, the authors performed CD8 depletion experiments during the priming phase? It would be better to do the depletion right before the tumor challenge, in particular for CD4 T cells which may be important during the priming phase.

We also attempted CD4 depletion as suggested by the reviewer. The problem is that using anti-CD4 antibodies to deplete CD4 T cells results not only in depletion of helper T cells (which may be helping the CD8 response) but also the regulatory T cells (which may be suppressing the CD8 response). Hence, CD4 depletion does not solve the problem. Indeed, our experiment gave us precisely the kind of ambiguous response reflective of the two-sides-of-the-knife activity of CD4 cells: CD4 depletion did have an influence on tumor growth in 2/5 mice but no influence whatsoever in the remaining 3/5 mice. We now show this observation in Supplementary Fig. 1b. This variability exists, in our thinking, because of variability of Treg activity in individual mice. Unfortunately, the reagents that can distinguish between the helper and the regulatory T cells do not exist at the moment. An indirect way to look for a contribution of CD4 cells was to look for the presence of MHC II neoepitopes within our active sequences. As described in the text on page 8 and shown in Supplementary Fig. 1 and Supplementary Table 1, we could not find such epitopes.

It seems quite odd that tumor growth data of mice immunized with the wild-type non-mutated Ccdc85cWT peptide (Fig. 2a) is displayed here. It seems more obvious to display these data at the beginning in Fig. 1b. Instead, this figure could start with the evidence showing that control immunization with Alms1MUT, does not lead to tumor rejection because this is the control used in the experiments described in Figure 2b-d.

This has now been corrected and Fig. 1 now contains the relevant data.

Figure 2c helps to define the different tumor-infiltrating cells, importantly clusters 1 and 5 which are further analyzed. How can immunization with another neoepitope such as Alms1MUT LYLDKSDTTV, which has a higher MHC-I affinity, not induce comparable responses

THAT is the point of our primary discovery as published in Duan et al. 2014, Ebrahimi-Nik et al. 2019, Brennick et al. 2021: that higher affinity neoepitopes are INEFFECTIVE at generating tumor immunity. As we have argued there, this situation is different from T cell responses to viral

antigens because neopeptides are altered self-antigens, and it is most likely that the high affinity TCRs to these have already been deleted.

In the experimental setting that does not involve immunization, enhanced immunogenicity of the low MHC-I affinity neopeptide should be functionally demonstrated in terms of tumor growth. If the presence of the mutant neopeptide is relevant, tumor growth should be affected either spontaneously or after administration of checkpoint blockade immunotherapies such as anti PD1 and/or CTLA-4. Depletion experiments should also be performed to address the contribution of CD8 and CD4 T cell populations.

We have now included these data in Supplementary Fig. 4b (pasted below).

Sup. Fig. 4. b. BALB/c mice were tumor challenged with either REV or MUT cell line. Each line represents tumor growth in a single mouse. On the right panel, total Area Under the Curve (AUC) scores for REV and MUT are plotted. Each bar shows the average total AUC score for the indicated group. Error bars represent standard deviation (SD). P values were calculated using t-test.

This manuscript would greatly improve by a robust analysis of Ccdc85cMUT-specific CD8 T cell responses. Authors should quantify and phenotypically characterize CD8 T cell responses specific for both wild-type (T)YIRPLETKVK- and mutated (T)YIRPFETKVK epitopes. To this end, they could perform flow cytometry-based analyses, such as MHC-multimer staining and/or intracellular cytokine staining after ex vivo peptide stimulation.

We fully appreciate the validity of this comment. That being said, known methodologies for characterizing antigen-specific T cells are not available to us precisely because the low affinity of the peptides for MHC I makes it impossible to generate tetramers for them. We now say so on page 6. We are in the process of developing new tools specifically for the low-affinity neopeptides, but they are some distance away. Please note that we have already demonstrated in Fig. 3C that clonally expanded TILs harvested from MUT1 and MUT2 but not from the REV tumor show significant expansion of genes associated with cytotoxicity (including Granzyme B and perforin), TCR engagement (various Nr4 genes) and other effector functions (interferon γ and others).

Along the different figures, statistical analysis is not informed for all relevant comparisons. The number of individual experiments performed for each analysis should be described.

Please note that the legend to Fig. 1d clearly states, "Error bars represent standard deviation (SD). *P* values were calculated using 1-way ANOVA test adjusted for multiple comparisons." Also, we have further analyzed the hierarchy shown in Fig. 3a and have added further statistical validation (as new Supplementary Fig. 6). We have also added in text on page 12, while discussing Fig. 3a upper panel, "with 97% confidence in the hierarchy edge/branch from Rev on one side to MUT1 and MUT2 on the other (see full details in Supplementary Fig. 6)". Please also note that the *P* values for Figure 3b and 3c are mentioned in the main text of the manuscript only to avoid having an extra-long legend for Fig 3. However, we can move the *P* values to the legend per the reviewer's preference.

Minor comments

The title does not quite reflect the main findings of the manuscript. It is not needed to remark a technical aspect of the study. In addition, to name neoepitopes with low affinity for MHC-class-I molecules as "non-MHC binders" seems misleading.

We have changed the title to: "Reversion analysis reveals the immunogenicity in vivo of a poorly MHC I-binding cancer neoepitope"

It is not possible to assure that the 18-mer is presented through cross-presentation by BMDCs. Alternatively, this peptide may potentially be degraded extracellularly generating shorter peptides that can be loaded onto MHC-I molecules, bypassing antigen cross-presentation. This concern is particularly latent, when using relatively high concentrations (100 μ M) of the peptide, as inferred by the authors' previous publication.

The point is well taken; however, we have extensively tested long peptides (even at high concentrations) and have never found them to be presentable by non-APCs. As an example, we have tested a number of long peptides for presentation by RMA-S cells, and they have consistently failed to do so (unpublished). We must add that we always use peptides of high purity (>98%).

REVIEWER COMMENTS

Reviewer #1 (Remarks to the Author):

The authors have adequately addressed the concerns in the initial review. There are no significant issues to address. This manuscript remains very timely and important to the field.

Reviewer #2 (Remarks to the Author):

No further comments.

Reviewer #3 (Remarks to the Author):

I consider that the revised version of the present manuscript is improved and addressed some of my comments. However, important concerns remain still unaddressed. First, as suggested in the first revision separating tumor curves from groups of animals immunized with the different truncated peptides would allow to analyze the antitumor activity of each peptide. Following the respective statistical comparisons, potentially interesting additional information could be obtained. Second, experiments involving CD8 and CD4 T cell depletions prior to tumor challenge (rather than the priming) were neither included nor discussed. The authors included an experiment with CD4 T cell depletion before the priming, which seems to show less protection (was the AUC analysis performed? is this significant?). Third, the manuscript would greatly improve by analysis of CD8 T cell responses specific for both wild-type (T)YIRPLETKVK- and mutated (T)YIRPFETKVK epitopes. I understand that preparing MHC-I multimers for such low affinity peptide is challenging. However, ex vivo stimulation with cognate peptides followed by intracellular IFN-gamma staining is a standard and affordable approach. Since you can control the concentration of the peptides (nM- μ M) added for the ex vivo stimulation, this technique can work even for detecting CD8 T cells specific for low affinity peptides.

Minor comment: It would have been very helpful for the reviewers if the authors had highlighted (with color or track-of-changes) all the modifications made to the revised manuscript.

REVISED Point-by-point response

MS# NCOMMS-20-47737A

Revised title: Reversion analysis reveals the immunogenicity in vivo of a poorly MHC I-binding cancer neoepitope

Ebrahimi-Nik et al.

Reviewer #1 (Remarks to the Author):

The authors have adequately addressed the concerns in the initial review. There are no significant issues to address. This manuscript remains very timely and important to the field.

No response necessary. We thank the reviewer.

Reviewer #2 (Remarks to the Author):

No further comments.

No response necessary. We thank the reviewer.

Reviewer #3 (Remarks to the Author):

I consider that the revised version of the present manuscript is improved and addressed some of my comments. However, important concerns remain still unaddressed.

First, as suggested in the first revision separating tumor curves from groups of animals immunized with the different truncated peptides would allow to analyze the antitumor activity of each peptide. Following the respective statistical comparisons, potentially interesting additional information could be obtained.

The relevant figure is now provided as a new Supplementary Fig. 1. The raw data for this supplementary figure are provided as a new raw data Excel file for Supplementary Fig. 1.

Second, experiments involving CD8 and CD4 T cell depletions prior to tumor challenge (rather than the priming) were neither included nor discussed.

We have now added the comment on page 8 that depletion of CD4 cells during the entire experiment including the priming as well as the effector phase, did not abrogate tumor rejection.

The authors included an experiment with CD4 T cell depletion before the priming, which seems to show less protection (was the AUC analysis performed? is this significant?).

Yes, AUC analysis was performed. We have now added the comment on page 8 as well as in the legend to Supplementary Fig.2b that the effect of CD4 depletion was not statistically significant.

Third, the manuscript would greatly improve by analysis of CD8 T cell responses specific for both wild-type (T)YIRPLETKVK- and mutated (T)YIRPFETKVK epitopes. I understand that preparing MHC-I multimers for such low affinity peptide is challenging. However, ex vivo stimulation with cognate peptides followed by intracellular IFN-gamma staining is a standard and affordable approach. Since you can control the concentration of the peptides (nM-uM) added for the ex vivo stimulation, this technique can work even for detecting CD8 T cells specific for low affinity peptides.

These data are now provided in a new Supplementary Fig. 3. We had done these experiments a long time ago but did not include them in the manuscript because the CD8 response detected against the mutant peptide was significant but weak. Regardless, the data are now provided.

Minor comment: It would have been very helpful for the reviewers if the authors had highlighted (with color or track-of-changes) all the modifications made to the revised manuscript.

We had indeed provided a track-edited version for the convenience of the reviewers. We are not sure why that version was not available to the reviewer. In this revision, we are highlighting in yellow, the changes made.

END of DOCUMENT

REVIEWERS' COMMENTS

Reviewer #3 (Remarks to the Author):

The authors have addressed my major concerns